# Dullard-mediated Smad1/5/8 inhibition controls mouse cardiac neural crest cells condensation and outflow tract septation

Jean-François Darrigrand[1], Mariana Valente[2], Glenda Comai[3], Pauline Martinez[1], Maxime Petit[4], Ryuichi Nishinakamura[5], Daniel S Osorio[6], Gilles Renault[7], Carmen Marchiol[7], Vanessa Ribes[8]*, Bruno Cadot[1]*

[1]INSERM - Sorbonne Université UMR974 - Center for Research in Myology, Paris, France; [2]Cellular, Molecular, and Physiological Mechanisms of Heart Failure team, Paris-Cardiovascular Research Center (PARCC), European Georges Pompidou Hospital (HEGP), INSERM U970, F-75737, Paris, France; [3]Stem Cells and Development, Department of Developmental & Stem Cell Biology, CNRS UMR 3738, Institut Pasteur, Paris, France; [4]Unité Lymphopoïèse – INSERM U1223, Institut Pasteur, Paris, France; [5]Institute of Molecular Embryology and Genetics, Kumamoto University, Kumamoto, Japan; [6]Cytoskeletal Dynamics Lab, Institute for Molecular and Cellular Biology, Instituto de Investigação e Inovação em Saúde, Universidade do Porto, Porto, Portugal; [7]Université de Paris, Institut Cochin, INSERM, CNRS, Paris, France; [8]Universite de Paris, Institut Jacques Monod, CNRS, Paris, France

**\*For correspondence:**
vanessa.ribes@ijm.fr (VR);
cadotbruno@gmail.com (BC)

**Competing interests:** The authors declare that no competing interests exist.

**Abstract** The establishment of separated pulmonary and systemic circulation in vertebrates, via cardiac outflow tract (OFT) septation, is a sensitive developmental process accounting for 10% of all congenital anomalies. Neural Crest Cells (NCC) colonising the heart condensate along the primitive endocardial tube and force its scission into two tubes. Here, we show that NCC aggregation progressively decreases along the OFT distal-proximal axis following a BMP signalling gradient. Dullard, a nuclear phosphatase, tunes the BMP gradient amplitude and prevents NCC premature condensation. Dullard maintains transcriptional programs providing NCC with mesenchymal traits. It attenuates the expression of the aggregation factor *Sema3c* and conversely promotes that of the epithelial-mesenchymal transition driver *Twist1*. Altogether, Dullard-mediated fine-tuning of BMP signalling ensures the timed and progressive zipper-like closure of the OFT by the NCC and prevents the formation of a heart carrying the congenital abnormalities defining the tetralogy of Fallot.

## Introduction

The heart outflow tract (OFT) is an embryonic structure which ensures the connection between the muscular heart chambers and the embryonic vascular network. Initially, forming a solitary tube called truncus arteriosus, it gets progressively remodelled into two tubes which give rise to the aortic (Ao) and pulmonary (Pa) arteries (*Brickner et al., 2000*; *Figure 1A*). This remodelling stands as one of the most sensitive processes during heart morphogenesis. As such, faulty septation of the OFT represents 30% of all congenital heart diseases, with poor clinical prognosis due to improper mixing of oxygenated and deoxygenated blood. This thus calls for a better understanding of the cellular and molecular cues by which the OFT gets septated during development.

Morphogenesis of the OFT is orchestrated in time and space by cross-interaction between several cell types including the myocardial progenitors of the second heart field (SHF), the endocardial cells

**Figure 1.** Dullard acts as a Smad1/5/8 activity inhibitor in cardiac NCC. (A) Ai. Schematic representation of the migration routes the cardiac NCC (green) have taken to reach the heart region (red) in a E10.5 mouse embryo. Aii. Schematics of the embryonic heart at E11.5 showing the distal-proximal axis of the OFT. Aiii. Schematic representation of transverse sections through the OFT showing discrete stages of NCC condensation and endocardium septation along the OFT distal-proximal axis. (B) Pecam and GFP immunolabelling and DAPI staining on transverse sections throughout the medial OFT of E11.5 $Wnt1^{Cre}$ or $Pax3^{Cre}$; $Dullard^{flox/+}$; $Rosa26^{mTmG}$ embryos. (C) Normalized expression levels of *Dullard* assayed by q-RT-PCR on single cells isolated after immuno-marking endothelial CD31+ cells from E11.5 $Wnt1^{Cre}$; $Dullard^{flox/+}$ and $Wnt1^{Cre}$; $Dullard^{flox/flox}$; $Rosa26^{mTmG}$ hearts (dots: value for a single cell; boxplot: mean ± s.e.m.). The primers used to amplify *Dullard* specifically binds to exons 2 and 3, which are excised by the *Cre* recombinase. (D) *Dullard* mRNA distribution detected using RNAscope probes, in transverse sections of E11.5 control and mutant OFTs, assessed by RNAscope. Dullard mRNA levels were significantly reduced in mutant cardiac cushions compared to controls; however, mRNA signals were still detected given the binding of Z pair probes to non-recombined exons 5 to 8 and UTR region. (E) Ei. Schematics of E11.5 heart showing the position of the transverse sections used to quantify the levels of the phosphorylated forms of Smad1/5/8 in iii. Eii. Immunolabelling for P-Smad1/5/8 and GFP, and DAPI staining on transverse sections across the OFT at three distinct distal-proximal levels in E11.5 embryos with the indicated genotype. Eiii. Quantification of P-Smad1/5/8 levels in cardiac NCC along the OFT distal-proximal axis of E11.5 embryos with the indicated genotype (dots: values obtained on a given section; n > 4 embryos per genotype recovered from at least three liters; the black line is the linear regression, the coloured areas delineate the 95% confidence intervals, ***: p-value<0001 for a two-way Anova statistical test). (F) *Msx2* and *Id2* mRNA distribution detected using RNAscope probes (grey) and immunostaining of GFP (green) in transverse sections of E11.5 control and mutant OFTs (n = 2 embryos). *On all A-F panels:* green dotted lines delineate the area colonised by cardiac NCC. Ao: aortic artery, Pa: pulmonary artery.

The online version of this article includes the following figure supplement(s) for figure 1:

**Figure supplement 1.** Dullard phosphatase is a negative regulator of BMP signalling in several mammalian cells.

(EC) delineating the OFT lumen, and the cardiac neural crest cells (cardiac NCC) (*Kelly, 2012*; *Keyte and Hutson, 2012*; *Figure 1A*). Various genetic manipulations or ablation models have highlighted the predominant role of cardiac NCC in initiating and controlling OFT septation (*Bockman et al., 1987*; *Phillips et al., 2013*). Originally, cardiac NCC delaminate from the dorsal neural tube and migrate through the pharyngeal mesoderm to reach the developing OFT (*Figure 1A*). There, they invade the two cardiac cushions, condense toward the endocardium and trigger its rupture, thereby inducing cardiac cushions fusion and creating the two great arteries (*Plein et al., 2015*; *Waldo et al., 1998*). The rupture of the endocardium is first detected in the regions of the OFT which are the most distal from the heart chambers. In mouse embryos this rupture initiates around 11.5 days of embryonic development (E11.5; *Figure 1A*) and then expands progressively to more proximal levels. In parallel to these morphogenetic events, NCC differentiate into the vascular smooth muscles of the aortic arch (*Keyte and Hutson, 2012*) and also contribute to the arterial valves (*Odelin et al., 2018*).

Intense investigations to identify the molecular cues controlling the stereotyped behaviour and differentiation of cardiac NCC in the OFT have established the importance of the Bone Morphogenic Proteins (BMP), secreted by the outlying myocardium cells from E8.75 onwards (*Danesh et al., 2009*; *Jiao et al., 2003*; *Liu et al., 2004*; *McCulley et al., 2008*). Indeed, ablation of the BMP receptor Bmpr1a, ablation of the key downstream transcriptional effector Smad4, or forced expression of the BMP signalling antagonist Smad7 within the NCC lineage all lead to the formation of hypoplastic cushions, a shorter and non-septated OFT, thus phenocopying cardiac NCC ablation experiments (*Jia et al., 2007*; *Stottmann et al., 2004*; *Tang et al., 2010*). Knock-out of the ligand BMP4 from the myocardium similarly prevents OFT septation (*Liu et al., 2004*). However, little is known about the cardiac NCC behaviour and molecular cascades triggered by BMP signalling and responsible for the cardiac NCC mediated OFT septation.

To gain insights into these molecular cascades, we decided to dissect the role of Dullard (Ctdnep1), a perinuclear phosphatase that functions as a negative intracellular BMP inhibitor, during OFT morphogenesis (*Sakaguchi et al., 2013*; *Urrutia et al., 2016*; *Sardi et al., 2019*). In the canonical BMP signalling cascade, binding of BMP ligands to their transmembrane receptors leads to the phosphorylation of the transcription factors Smad1/5/8 which translocate to the nucleus and modify the transcriptional landscape of targeted cells (*Bruce and Sapkota, 2012*). Dullard stands out as one of the few cytoplasmic modulators of this phosphorylation step, which also includes PP1A, PP2B, the inhibitory Smads 6 and 7 and the Ubiquitin degradation pathway (*Bruce and Sapkota, 2012*). The Dullard protein is evolutionary conserved from yeast to mammals and expressed in many embryonic tissues, including the developing neural tube and neural crest cells (*Sakaguchi et al., 2013*; *Satow et al., 2006*; *Tanaka et al., 2013*; *Urrutia et al., 2016*; *Figure 1D*). Several pieces of evidence from *Drosophila*, xenopus, and mouse embryos indicate that this enzyme dampens Smad1/5/8 phosphorylation levels upon BMP stimulation (*Sakaguchi et al., 2013*; *Satow et al., 2006*; *Urrutia et al., 2016*). However, this activity is likely to be tissue specific, as depleting Dullard in gastrulating mouse embryos or later in the limb bud mesenchyme did not impair BMP signalling, while its depletion in the mouse embryonic kidney led to an elevated BMP response (*Sakaguchi et al., 2013*; *Tanaka et al., 2013*; *Hayata et al., 2015*). Regardless of the effect on BMP signalling, Dullard appears as a key regulator of various morphogenetic events regulating the elaboration of embryonic tissues. Early in development, it is required for the expansion of extraembryonic tissues, and later on, it prevents cell death by apoptosis in kidney nephrons or favours the ossification of limb bones (*Sakaguchi et al., 2013*; *Tanaka et al., 2013*; *Hayata et al., 2015*).

We showed here that deletion of *Dullard* in the cardiac NCC increases Smad1/5/8 activity, leading to premature and asymmetric septation of the OFT and pulmonary artery closure. BMP overactivation in the cardiac NCC occurs concurrently with the downregulation of mesenchymal markers (*Snai2*, *Twist1*, *Rac1*, *Mmp14* and *Cdh2*) and upregulation of *Sema3c*, which is associated with premature cardiac NCC condensation to the endocardium. Our data converge to a model whereby graded BMP activity, *Sema3c* expression and cardiac NCC condensation along the OFT axis set the tempo of OFT septation from its distal to its proximal regions. Hence, our findings reveal that fine tuning of BMP signalling levels in cardiac NCC orchestrates OFT septation in time and space.

## Results

### *Dullard* deletion triggers hyperactivation of BMP intracellular signalling in cardiac NCC

In order to ablate Dullard in cardiac NCC, we crossed mice carrying floxed alleles of *Dullard* with mice expressing the Cre recombinase from the *Pax3* locus or thanks to *Wnt1* enhancer (*Danielian et al., 1998*; *Engleka et al., 2005*; *Sakaguchi et al., 2013*). Cell lineage tracing was achieved by using a ubiquitous double-fluorescent Cre reporter allele, *Rosa26^{mTmG}*, in which Cre-mediated recombination labels the cells with membrane-targeted GFP (*Muzumdar et al., 2007*). The pattern of cell recombination in the cardiac cushions of E11.5 control embryos carrying either Cre driver matched with the pattern of colonising cardiac NCC described by previous lineage analyses (*Figure 1B*; *Brown et al., 2001*; *Jiang et al., 2000*). RT-qPCR on single cells isolated by Fluorescence-activated cell sorting (FACS) from dissected OFT and RNAscope in situ hybridization on histological sections were used to monitor *Dullard* expression and validate its deletion on GFP$^+$ cells upon Cre recombination of *Dullard* flox alleles (*Figure 1C,D*). At E11.5, *Dullard* was ubiquitously expressed in all OFT layers of control embryos. In recombined *Wnt1^{Cre}*; *Dullard^{flox/flox}*; *Rosa26^{mTmG}* embryos, the cardiac NCC displayed a strong reduction in *Dullard* levels compared to control littermates, while the surrounding tissues remained *Dullard* positive. Strikingly, in these mutants, the NCC formed a unique mass at the distal part of the OFT, while two distinct NCC cushions were present in the control embryos (*Figure 1D*), indicating that Dullard regulates the spatial organization of NCC in the OFT (see below).

We next assessed the relationship between Dullard and the activity of the intracellular effectors of BMP signalling, that is the Smad1/5/8 transcription factors, in mammalian cells. As previously shown (*Satow et al., 2006*), Dullard overexpression in the myogenic cell line C2C12 strongly decreased the levels of phosphorylated Smad1/5/8 induced by BMP2 treatment (*Figure 1—figure supplement 1A*). In addition, by generating a version of Dullard carrying a phosphatase dead domain, we showed that the role of Dullard as negative modulator of BMP signaling relied on its phosphatase activity (*Figure 1—figure supplement 1A*). Accordingly, *Dullard* deletion in cardiac NCC was sufficient to double the levels of P-Smad1/5/8 within the NCC whatever their position along the distal-proximal OFT axis of E11.5 hearts (*Figure 1E*, *Figure 1—figure supplement 1B*). To confirm this result, we also looked at the distribution and/or the levels of expression of several well-established BMP signalling pathway downstream targets, namely *Id1*, *Id2*, *Msx1* and *Msx2* (*Figure 1F*, *Figure 1—figure supplement 1C*; *Miyazono et al., 2005*). Quantitative RT-qPCR on isolated E11.5 NCC showed that the levels of *Id1* transcripts were significantly elevated by Dullard loss and that *Msx1* and *Msx2* expression range was shifted towards higher values in mutants compared to control embryos (*Figure 1—figure supplement 1C*). In agreement, the amount of *Msx2* transcripts detected by in situ hybridisation within the NCC in the vicinity of the aorta was greater in mutants compared to control embryos, so was that of *Id2* transcripts present throughout all cardiac NCC (*Figure 1F*). This further indicates that in the cardiac NCC lineage, Dullard acts as a BMP intracellular signalling inhibitor. It is worth mentioning that Dullard was also required to dampen the levels of activated Smad1/5/8 in other cell types, including myogenic cells (*Figure 1—figure supplement 1D*). Furthermore, while Dullard has also been shown to decrease the levels of phosphorylation of Smads acting downstream of the TGF$\beta$ in bone precursors (*Hayata et al., 2015*), the modulation of Dullard activity did not alter the phosphorylation state of one of these Smads, namely Smad2, in cardiac NCC (*Figure 1—figure supplement 1E*).

Remarkably, in both control and mutant contexts, P-Smad1/5/8 levels were more elevated distally than proximally (*Figure 1Eiii*, *Figure 1—figure supplement 1Bii*), indicating that BMP signalling elicits a graded response in cardiac NCC, which declines as they colonise more proximal OFT areas. Altogether, our results show that Dullard is required in cardiac NCC to dampen the magnitude of the BMP signalling gradient along the proximo-distal axis of the OFT, but is not required for its establishment, which is still observed in mutant embryos.

## Dullard deletion in cardiac NCC leads to the emergence of heart abnormalities present in Fallot's tetralogy

Given that cardiac NCC control OFT septation (*Bockman et al., 1987*; *Phillips et al., 2013*; *Plein et al., 2015*), we next sought to examine the morphology of the OFT in control and *Dullard* mutants. No gross morphological defects were detected in E10.5 Dullard mutant OFTs (*Figure 2—figure supplement 1A*). Notably, the endocardium morphology, the thickness and position of the myocardium, the NCC distribution along the outflow tract, were comparable in control and mutant embryos (*Figure 2—figure supplement 1A*). Strikingly, 24 hr later, severe and penetrant morphological OFT defects were observed in Dullard mutant embryos (*Figure 2A, B*, *Figure 2—figure supplement 1B*). To characterize these defects, we first analyzed E11.5–12 hearts labelled for the arterial marker Pecam using 3D lightsheet and confocal microscopy (*Figure 2A,B*, *Figure 2—figure supplement 1B*, *Videos 1* and *2*). At distal levels, the OFT of control embryos displayed symmetrical septation with two great arteries of similar size (*Figure 2Ai*,Bi, *Figure 2—figure supplement 1Bi - Video 1*). In contrast, *Pax3^{Cre}* or *Wnt1^{Cre}*; *Dullard^{flox/flox}* embryos exhibited an asymmetric breakdown of the endocardium on the pulmonary side with obstruction of the pulmonary artery (Pa) (*Figure 2Aii*,Biv, *Figure 2—figure supplement 1Biv - Video 2*). At more medial levels in control embryos, the pulmonary pole of the endocardium was still connected to its aortic pole (*Figure 2Bii*, *Figure 2—figure supplement 1Bii*). This pole was also attached to the presumptive pulmonary valve intercalated-cushion (PV-IC), a cell cluster recognisable by faint levels of endothelial markers such as Pecam (*Mifflin et al., 2018*). Conversely, the aortic and pulmonary poles of the endocardium were prematurely septated and the NCC cushions were fused in the medial portion of the mutant OFT (*Figure 2Bv*, *Figure 1—figure supplement 1Bv*). Similarly to the observations made at distal levels, pulmonary endocardium cells were aggregated together failing to delineate a lumen. Furthermore, NCC often intervened between the residual pulmonary endocardium cells and the presumptive pulmonary valve intercalated-cushion (PV-IC) (*Figure 2Bv*). At proximal regions, the shape of the OFT endocardium and the surrounding NCC cushions were similar in both mutant and control embryos (*Figure 2A,Biii,vi*, *Figure 2—figure supplement 1Biii,vi*). We also checked for the state of the peripheral sheet of myocardium composing E11.5 control and Dullard mutant OFTs by immunolabelling the myosin heavy chain or the transcription factor Islet1/2 (Isl1/2) (*Figure 2—figure supplement 1C,D*). No significant differences could be detected between the control and mutant OFTs.

In order to evaluate the physiological impact of these OFT defects on cardiac function, we first attempted to harvest older embryos to conduct a histological characterisation of their hearts. The premature death of *Pax3^{Cre}* or *Wnt1^{Cre}*; *Dullard^{flox/flox}* embryos after E12.5 complicated this task (*Figure 2C*). Nevertheless, in the handful of Dullard mutant embryos collected alive at E14.5 or E18.5, we consistently observed hearts with an interventricular septum (compare *Figure 2Dii*,iv with 2Di,iii), a hypertrophy of their right ventricle compared to their left one (compare *Figure 2Div* with 2Diii), a wide aorta connected to both ventricles (*Figure 2Dvi*, E) and pulmonary stenosis (compare *Figure 2Dvi* with 2Dv, asterisk in E). These are the four major heart morphological traits defining the tetralogy of Fallot (TOF)'s condition (*Neeb et al., 2013*). Second, we imaged the blood flow passing through the abdominal artery of E11.75 *Wnt1^{Cre}*; *Dullard^{flox/flox}* and *Wnt1^{Cre}*; *Dullard^{flox/+}* embryos using doppler ultrasound (*Figure 2F*; see Material and methods section; *Nomura-Kitabayashi et al., 2009*). Several hemodynamic parameters were affected in Dullard mutants compared to controls. Notably, the systolic velocity peaked at a lower level in mutants compared to controls, suggesting a compromised blood ejection from the heart (*Figure 2Fi*). Conversely, the heart relaxation phase was less affected as no differences in the diastolic velocity were detected (*Figure 2Fii*). Overall, the blood flow was weaker in the mutants than in controls, as indicated by a decrease in the mean of the velocity time integral (VTI) (*Figure 2Fiii*).

## Dullard prevents the premature condensation of cardiac NCC

We next wanted to further investigate the cellular mechanisms by which Dullard in NCC ensures OFT septation and started evaluating the migrative, proliferation and death status of cardiac NCC (*Figure 3A*, *Figure 3—figure supplement 1*, *Videos 1–4*). Whole mount immunostaining and 3D-reconstructions revealed that GFP+ cardiac NCC reached similar OFT levels in E11.5 control and

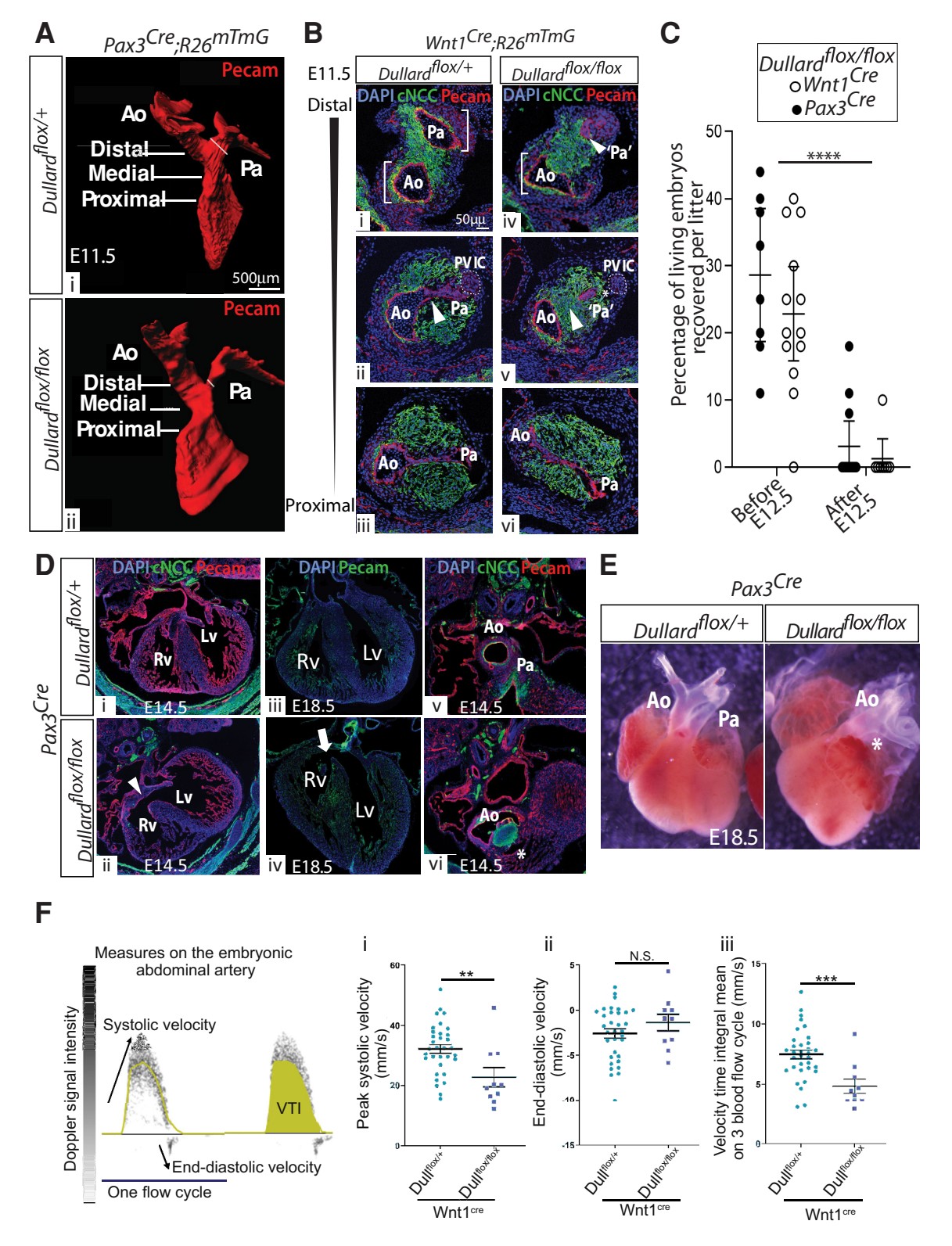

**Figure 2.** *Dullard* deletion in cardiac NCC causes asymmetric and premature OFT septation similar to Fallot's tetralogy. (**A**) Three-dimensional rendering of the Pecam⁺ endocardium of E12 *Pax3^Cre^; Dullard^flox/+^* and *Pax3^Cre^; Dullard^flox/flox^* embryos after 3Disco clearing and lightsheet acquisition (n = 3 per genotype). The fine oblique white line marks the Pa width. The OFT levels along its distal-proximal axis analyzed in B are also indicated. (**B**) Immunolabelling for Pecam (red), GFP (green) and DAPI (blue) on transverse sections along the distal-proximal axis of the OFT in E11.5 embryos with

*Figure 2 continued on next page*

*Figure 2 continued*

the indicated genotypes (n > 10 embryos collected from more than three liters). Brackets in i and iv highlight the symmetric and asymmetric Ao and Pa poles in control and mutant embryos, respectively. Arrowheads in ii and v point at the unruptured and ruptured endocardium in control and mutant embryos, respectively. (**C**) Percentage of living Dullard mutant embryos before E12.5 and after E12.5, carrying the indicated Cre driver. (**D**) Immunolabelling for Pecam, GFP and DAPI staining on sections through the hearts of E14.5 (i,ii,v,vi) and E18.5 (iii,iv) embryos with the indicated genotypes (n = 2 embryos per genotype). Arrowheads in ii and arrow in iv point at a septation defect, the star in vi indicates the lack of Pa. (**E**) Whole dissected E18.5 hearts coming from embryos with the indicated genotype (n = 2 per genotype). (**F**) Two cycles of blood flow measured at the level of the abdominal artery of E11.75 control embryos and indication of the parameters analysed. VTI: velocity time integral. Parameters (i-iii) of the blood flow velocity measured in the abdominal artery of E11.75 control (turquoise dots) and *Wnt1Cre; Dullard^flox/flox* embryos (purple squares)(dots and squares: mean of two to five measures obtained on a single embryo, bars: mean ± s.e.m; differences evaluated using a Mann-Whitney test: N.S. non-significant, *: p<0.05, **: p<0.01, ***: p<0.001). i. peak systolic velocity, ii. end-diastolic velocity, iii. mean of three velocity time integrals (n = 10 mutants and n = 32 controls). Ao: aortic artery, Pa: pulmonary artery, PV IC: pulmonary valve intercalated-cushion.

The online version of this article includes the following figure supplement(s) for figure 2:

**Figure supplement 1.** Morphological defects in Dullard mutants become striking from E11.5 onwards.

*Dullard* mutants, showing that Dullard is not required for NCC colonisation of the OFT (*Figure 3A*, *Videos 1–4*). Similarly, quantification of cell proliferation and apoptosis on tissue sections, using antibodies raised against the phosphorylated form of histone H3 and the cleaved version of Caspase three respectively, indicated that Dullard does not control the proliferation nor the survival of cardiac NCC (*Figure 3—figure supplement 1Ai-iv*). In agreement with these observations, the total number of GFP$^+$ cells colonising the OFT in mutant embryos was not significantly different from that found in controls (*Figure 3—figure supplement 1Av, vi*).

Finally, we wondered whether the morphogenetic defects of the mutant OFT could stem from differences in cell-cell arrangements, looking at the position and orientation of NCC and endocardial cell nuclei (*Figure 3B–D*, *Figure 3—figure supplement 1B–C*). The orientation of the cardiac NCC nuclei relative to the endocardium appeared spatially regulated along the proximal-distal axis of the OFT, in both mutant and control hearts (*Figure 3B,C*, *Figure 3—figure supplement 1B*). In controls, NCC perpendicular to the endocardium could be found at distal levels, while at proximal levels no orientation preference could be assigned (blue dashes in *Figure 3B*, *Figure 3—figure supplement 1B*). Strikingly, in *Wnt1^Cre; Dullard^flox/flox* OFTs the perpendicular orientation was more widely observed at medial levels than in control OFT (blue dashes in *Figure 3B*, *Figure 3—figure supplement 1B*). Moreover, quantification of the shortest distance between adjacent cardiac NCC nuclei indicated that in E11.5 control hearts NCC condensation was also variable along the distal-proximal axis of the OFT (*Figure 3C,Di*). Cells were closer to each other at distal levels than in proximal regions. This progression of NCC condensation along the OFT axis was impaired in *Dullard* mutants, whereby mutant NCC prematurely condensed within the medial region of the OFT (*Figure 3C, Di*). Finally, the position of NCC to the endocardium was variable along the OFT axis of control embryos with NCC being closer to this epithelium at distal levels than at proximal levels (*Figure 3C,Dii*). In the mutants, NCC were in a closer vicinity of the endocardium than control cells, so that in medial levels they displayed traits of cells normally found at distal levels in control hearts (*Figure 3C,Dii*). In agreement with these data, the OFT area was reduced in mutants and remained more constant along the distal to proximal axis (*Figure 3—figure supplement 1C*).

Taken together *Figures 2* and *3* data demonstrate that *Dullard* stands as a key modulator of NCC behaviour dynamics in the heart and hence of OFT septation. It precipitates NCC condensation, and thereby leads to the

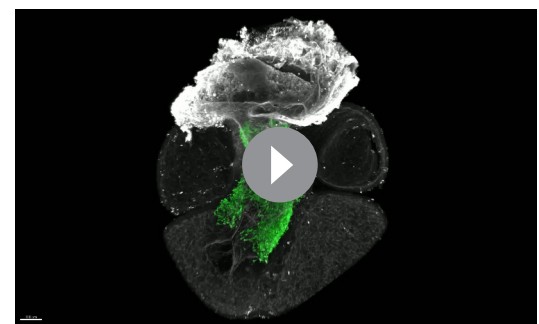

**Video 1.** Three-dimensional rendering of cardiac NCC (green) over Pecam (white) after BABB clearing and Lightsheet acquisition of *Pax3^Cre; Dullard^flox/+; Rosa26^mTmG* and *Pax3^Cre; Dullard^flox/flox; Rosa26^mTmG* E12 embryos. No defect in the OFT colonisation of mutant cardiac NCC is observed.
https://elifesciences.org/articles/50325#video1

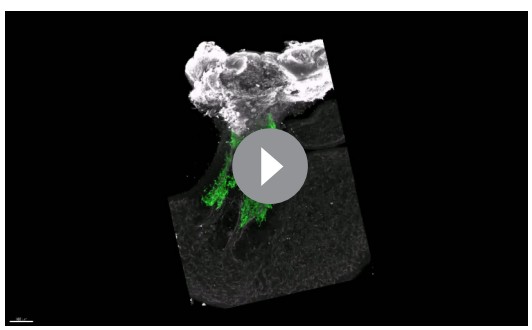

**Video 2.** Three-dimensional rendering of cardiac NCC (green) over Pecam (white) after BABB clearing and Lightsheet acquisition of *Pax3^{Cre}; Dullard^{flox/+}; Rosa26^{mTmG}* and *Pax3^{Cre}; Dullard^{flox/flox}; Rosa26^{mTmG}* E12 embryos. No defect in the OFT colonisation of mutant cardiac NCC is observed.
https://elifesciences.org/articles/50325#video2

premature breakage of the endocardium and obstruction of the pulmonary artery. This weakens the embryonic hemodynamics and compromises the living of Dullard mutant embryos (see discussion). Our data also brings further support to the idea that morphogenetic defects in the NCC-derived cushions stand as one possible cause of Fallot's tetralogy (*Neeb et al., 2013*).

## *Dullard* deletion in NCC mainly affects the transcriptional state of NCC

To decipher the molecular basis of the defective OFT remodeling observed in mutants, we microdissected *Wnt1^{Cre}; Rosa26^{mTmG}* E11.5 control and Dullard mutant heart OFTs and sorted the cardiac NCC (GFP$^+$) and endocardial cells (CD31$^+$; RFP$^+$) from the other OFT cell-types (CD31$^-$; RFP$^+$) (*Figure 4A*). We then performed single-cell RT-qPCR for 44 genes implicated in epithelial-mesenchymal transition (EMT), migration and/or specification of the different OFT progenitor subtypes (*Supplementary file 1*), and their expression levels were normalised to GAPDH and ActB (*Figure 4B*).

T-statistic Stochastic Neighbour Embedding (t-SNE) was first used to plot the distances existing between the 44 gene-based-transcriptomes of individual cells (*Figure 4C*). It revealed that the 44 chosen genes were sufficient to segregate the three isolated cell subtypes, GFP$^+$ NCC, the RFP$^+$; CD31$^+$endocardial cells and the other RFP$^+$;CD31$^-$ OFT cells, validating our approach. Unsupervised hierarchical clustering analysis of all cells refined this segregation and identified six distinct groups of OFT cells (*Figure 4D*). Importantly, some of these groups contained both control and *Dullard* mutant cells (Groups 1, 3, 5) meaning that their 44 gene-based-transcriptome was not drastically dependent on Dullard. Instead, the three other groups were enriched for cells with a given genotype (Groups 2, 4, 6), hence harboured a Dullard dependent transcriptional state. Importantly, most GFP$^+$ NCC were contained in the Groups 2, 4, 6, while the other groups were enriched for other cell types. For instance, the RFP$^+$;CD31$^+$ (*Flt1$^+$;Kdr$^+$;Nfatc$^+$;Tek$^+$*) endocardial cells and the RFP$^+$;CD31$^-$ (*Tcf21$^+$; Wt1$^+$*) epicardial cells defined the Groups 1 and 5, respectively. Group 3 contained *Isl1$^+$* and *Six2$^+$* cells coming from GFP$^+$ (NCC) and RFP$^+$;CD31$^-$ (including myocardial) lineages. Overall, it suggests that Dullard deletion in NCC mainly alters the transcriptional states cell-autonomously and modulates to a much lesser extent the surrounding cell types present in the developing E11.5 heart.

## Transcriptomic heterogeneity in non NCC-derived populations upon Dullard deficiency

We next focused on the transcriptomic variations operating in the distinct OFT specific cell populations, starting with the RFP$^+$;CD31$^+$endocardial cells and RFP$^+$;CD31$^-$ OFT cells (*Figure 5Ai–ii'*, *Figure 5—figure supplement 1*). Hierarchical clustering and two-dimensional visualisation of cells on diffusion maps indicated that in both cell types, some transcriptomic heterogeneity was found and distinct subpopulations could be isolated (*Figure 5Ai–ii'*, *Figure 5—figure supplement 1A*). All 5 RFP$^+$;CD31$^-$ subpopulations identified contained both control and mutant cells (*Figure 5Ai,i'*), sustaining the idea that Dullard loss in cardiac NCC does not impair the differentiation of the SHF-derived myocardium and smooth muscle, nor the differentiation of the epicardium (see also *Figure 2—figure supplement 1C*). Similarly, the vast majority of RFP$^+$;CD31$^+$ cells (four out of six subpopulations (sub-pops 2 to 5)) presented both mutant and control cells (*Figure 5Aii,ii'*, *Figure 5—figure supplement 1A*). The transcriptomic heterogeneity between these RFP$^+$;CD31$^+$ subpopulations was mild and these cells were all *Kdr$^+$*, *Foxc1$^+$*, *Nfatc^{high}*, *Flt1$^+$*, *Nrp1$^+$*, *Plxnd1* High, as expected for the endocardium (blue rectangles in *Figure 5—figure supplement 1A*). However, the subpopulation 1 of RFP$^+$;CD31$^+$ cells was enriched in control cells while the subpopulation six in

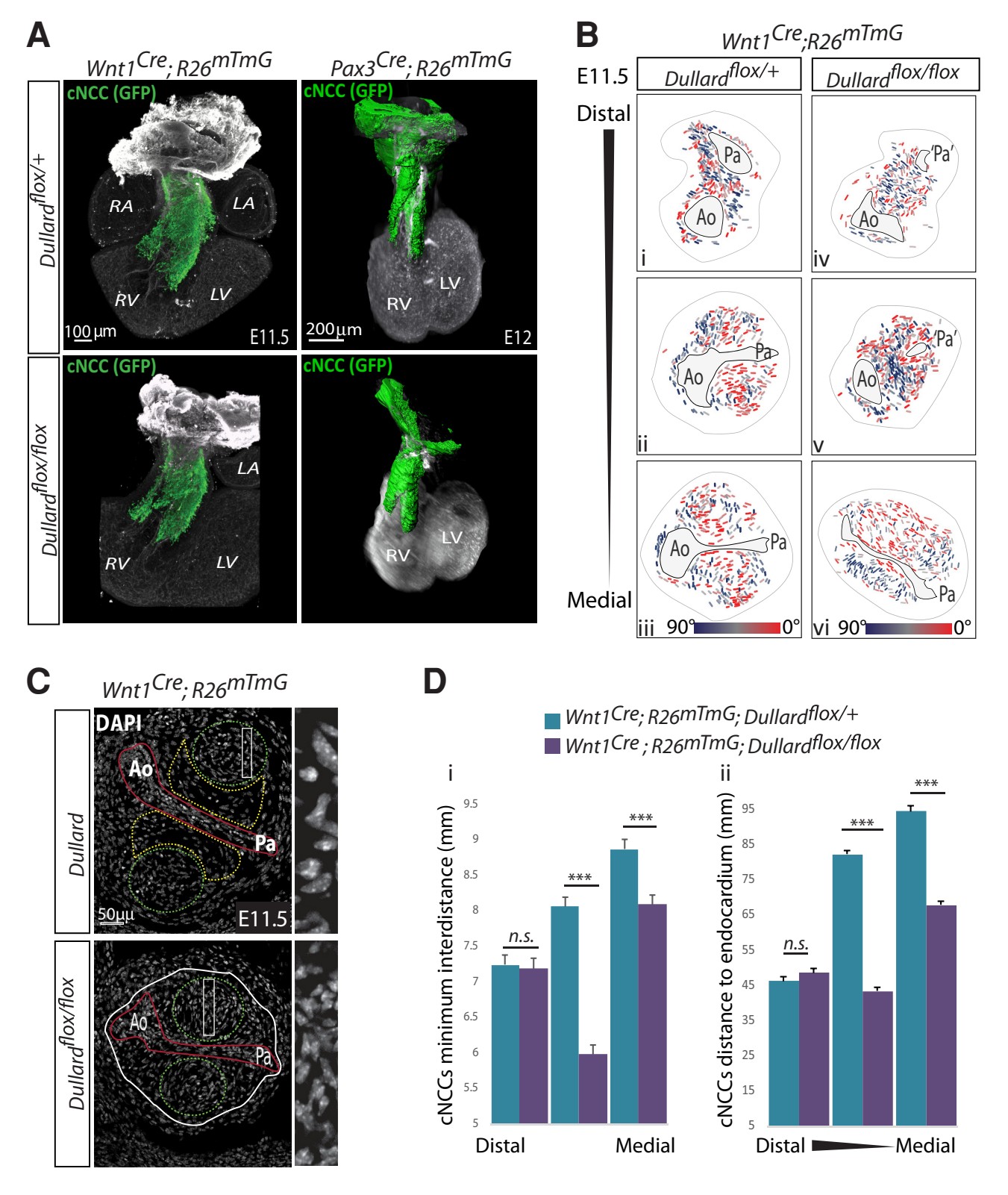

**Figure 3.** Dullard does not affect NCC migration, but prevents NCC premature condensation. (**A**) Three-dimensional rendering of cardiac NCC (green) over Pecam (white) after BABB clearing and confocal acquisition (*Wnt1cre* samples) or 3disco clearing and lightsheet microscopy (*Pax3cre* samples) of whole E11.5 hearts isolated from embryos with the indicated genotype (n = 2 per genotype). (**B**) Coloured coded orientation of the major axis of NCC cells relative to Ao-Pa axis colour-coded as indicated in the section shown in *Figure 2Bi–vi*. (**C**) DAPI staining on transverse sections through the medial
*Figure 3 continued on next page*

*Figure 3 continued*

part of the OFT of E11.5 embryos with the indicated genotype. Magnified regions on the right are indicated by white rectangles in NCC cushions. The entire OFT is circled with a while line. The endocardium is delineated in red, the condensed and round NCC in green, the loose and elongated NCC in yellow. (D) Minimum distances between NCCs (Di) and distances between NCCs and the endocardium (Dii) quantified along the distal-proximal axis of the OFT in E11.5 embryos with the indicated genotypes (n = 3 embryos from distinct liters were analyzed for each genotype and OFT level, bars: mean ±s.d.; ***: p-value<0.0001 for Student statistical t-test). Ao: Aorta; Pa: pulmonary artery; Lv: left ventricle; PV-IC: Pulmonary valve intercalated-cushion; Rv: right ventricle.

The online version of this article includes the following figure supplement(s) for figure 3:

**Figure supplement 1.** Morphogenetic defects in Dullard NCC mutants.

---

mutant cells (*Figure 5Aii*,ii'). From the genes that drive the segregation of these CD31$^+$ subpopulations (*Figure 5—figure supplement 1B*), very few of them were specifically induced or repressed in the subpopulation six or the subpopulation one compared to the others subpopulation (*Figure 5—figure supplement 1C*). Out of them stood *Twist1* and *Sox9*, which were enriched in the subpopulation 1. Accordingly, while few Twist1$^+$ cells could be immunolabelled within the endocardium of control embryos, these were almost undetectable in mutant embryos (*Figure 6A*). Twist1 being one of the epithelial-mesenchymal transition (EMT) drivers, it suggests that endocardial EMT is affected by the absence of Dullard in the NCC (see discussion).

## Dullard controls the mesenchymal transcriptional state of cardiac NCC

Given the key defects observed in NCC condensation in Dullard mutants, we then focused on the gene expression signature of GFP+ NCC. With similar approaches as above we could identify five GFP+ NCC subpopulations based on their gene expression signature (Sub Pops 1 to 5) (*Figure 5B*, *Figure 5—figure supplement 2A*), each of them containing an unbalanced ratio of mutant versus control cells (*Figure 5C,D*), suggesting that Dullard influences the fate of all NCC subtypes. The NCC subpopulation 1 was characterised by the expression of *Nfatc1, Tcf21, Postn (Periostin)* (*Figure 5—figure supplement 2A–C*), which are all expressed in the heart valves (*Acharya et al., 2011*; *Norris et al., 2008*; *Wu et al., 2011*), a heart structure that is also colonised by NCC (*Odelin et al., 2018*). The slight enrichment for control cells in this subpopulation raised the possibility that Dullard is required to favour the contribution of NCC to this structure (*Figure 5C*). The second subpopulation was defined by the predominant expression of cardiac progenitor markers *Tbx1, Six2, Gja1, Isl1*, and contained both control and mutant cardiac NCC (*Figure 5—figure supplement 2A–C*),

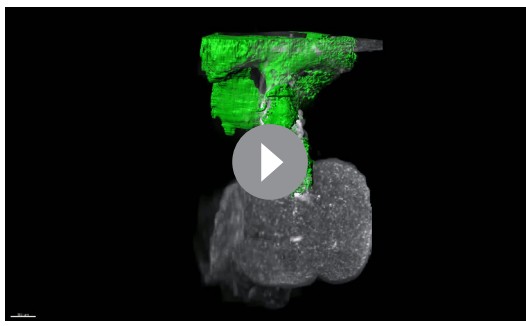

**Video 3.** Three-dimensional rendering of cardiac NCC (green isosurface) over Pecam (red isosurface) and Dapi (white) of *Wnt1$^{Cre}$; Dullard$^{flox/+}$; Rosa26$^{mTmG}$* and *Wnt1$^{Cre}$; Dullard$^{flox/flox}$; Rosa26$^{mTmG}$* E11.5 embryos. No defect in the OFT colonisation of mutant cardiac NCC is observed, and reduction of the pulmonary artery is visible in the mutant.

https://elifesciences.org/articles/50325#video3

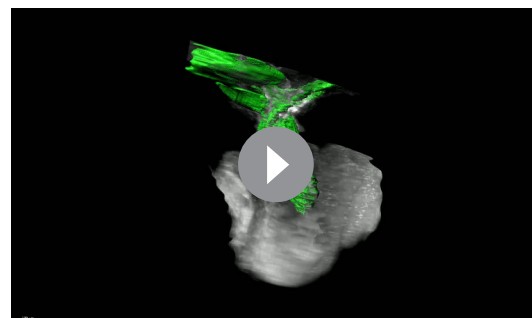

**Video 4.** Three-dimensional rendering of cardiac NCC (green isosurface) over Pecam (red isosurface) and Dapi (white) of *Wnt1$^{Cre}$; Dullard$^{flox/+}$; Rosa26$^{mTmG}$* and *Wnt1$^{Cre}$; Dullard$^{flox/flox}$; Rosa26$^{mTmG}$* E11.5 embryos. No defect in the OFT colonisation of mutant cardiac NCC is observed, and reduction of the pulmonary artery is visible in the mutant.

https://elifesciences.org/articles/50325#video4

indicating that Dullard is not required for the entry of NCC into the smooth muscle lineage (*Zhou et al., 2017*). This is in agreement with the distribution of Isl1 we observed in control and mutant embryos (*Figure 2—figure supplement 1C*). Conversely, the emergence of the subpopulation 3 was strictly dependent on Dullard, as this subpopulation barely contained mutant cells (*Figure 5C,D*, *Figure 5—figure supplement 2A*). Subpopulation 3 corresponded to NCC further differentiated toward the smooth lineage as defined by expression of *Myh11* and *Acta2* (*Huang et al., 2008*; *Figure 5D,E*). The subpopulations 4 and 5 were enriched for mutant cardiac NCC (*Figure 5C–E*, *Figure 5—figure supplement 2A*). The transcriptomic state of the cells in these two subpopulations diverged from that of the subpopulation 3 cells (*Figure 5C,E,D*). This divergence was greater for the subpopulation 5 than for the Ssubpopulation 4. Yet, the position of cells within the diffusion maps suggests a close relationship between these two subsets of cells which might represent two states of the differentiation path of *Dullard* deleted cardiac NCC (*Figure 5C*).

Strikingly, genes that were differentially expressed between subpopulation 3 and the subpopulation 4/5 encode for known regulators of cell adhesion and epithelial-mesenchymal transition. On the one hand, the levels of mesenchymal markers (*Snai2*, *Twist1*, *Cdh2*, *Mmp14*, *Rac1*) in both subpopulations 4 and 5 were lower than in the subpopulation 3. In control embryos, NCC expressing these markers were found near the endocardium, as demonstrated by an ISH for Snai2 or immunolabelling for Twist1 (*Figure 6A*). In mutants, these pro-epithelial-mesenchymal transition factors were barely detected in NCC. These data are in agreement with the NCC over-condensation phenotype observed in Dullard mutants nearby the endocardium. On the other hand, cells in subpopulations 4 and 5 displayed higher levels of the epithelial markers Cdh5 and of Sema3c compared to subpopulation 3 cells. In situ hybridization of Sema3c further confirmed these results with cellular resolution (*Figure 6B*, *Figure 6—figure supplement 1A,B*), showing that at distal OFT levels, *Sema3C* was expressed in a scattered fashion in NCC in cushions of control embryos, while it was found in almost all NCC in mutants. Importantly, we observed a graded decrease of *Sema3c* expression along the OFT axis in both control and mutant embryos. This was reminiscent of the BMP response and condensation gradients described previously (*Figures 1E* and *3D*) and in agreement with the established role of *Sema3c* in the regulation of cohesive/metastatic balance in several cancers (*Tamagnone, 2012*).

Altogether, our results show that Dullard prevents the establishment of an epithelial-like state and promotes the expression of pro-mesenchymal genes in NCC. In addition, it may also impair the epithelial mesenchymal transition of the endothelial cells (see discussion).

## Discussion

Our analysis of the functions of the Dullard phosphatase in cardiac NCC brings us to propose a model whereby a BMP-dependent gradient of cardiac NCC condensation would set the timing of cardiac cushions fusion and thereby the septation of the OFT into the aorta and pulmonary artery. In addition, the analogy between the phenotype of Dullard mutant mice and patients suffering from Fallot's tetralogy condition calls for a discussion on the cellular and molecular aetiology of this complex congenital heart disease.

First, we showed that a gradient of BMP activation exists in the cardiac NCC along the OFT axis and matches with a gradient of NCC condensation. The magnitude of this BMP gradient is under the control of the phosphatase *Dullard*. Yet, the establishment of this gradient is dependent on other mechanisms, as the gradient is still observed in absence of *Dullard* in the cardiac NCC. The establishment of this gradient is unlikely to result from a corresponding gradient of ligand that would diffuse from a localized distal source, as described in other contexts (*Bier and De Robertis, 2015*). In fact, BMP4 is homogeneously expressed in the myocardium throughout the entire length of the OFT and not solely at the distal level (*Jia et al., 2007*; *Jiao et al., 2003*; *Jones et al., 1991*; *Zhang et al., 2010*). Rather, intracellular signalling inhibitors could allow a temporal adaptation of cardiac NCC to BMP signals along with their migration towards proximal levels of the OFT. Prominent expression of *Smad6*, a BMP negative feedback effector, and of the diffusible inhibitor *Noggin*, are indeed observed in the OFT from E10.5 throughout the great arteries formation and thus represent promising candidates (*Choi et al., 2007*; *Galvin et al., 2000*).

Secondly, several lines of evidence support the idea that the elevation of BMP signalling in Dullard mutants underpins the premature condensation of cardiac NCC and the strong pulmonary artery

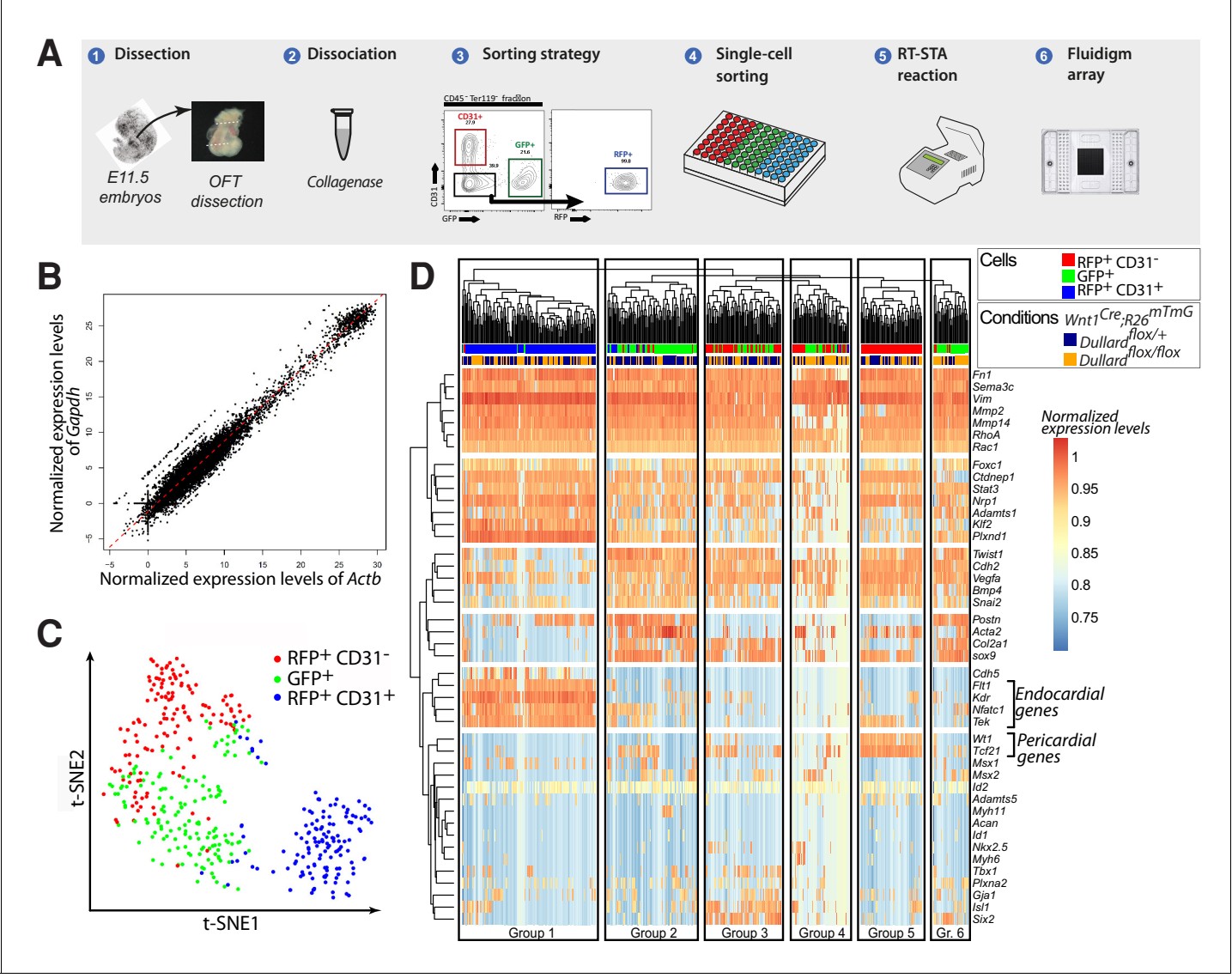

**Figure 4.** Single-cell transcriptional analyses of all OFT cells at E11.5. (**A**) Experimental steps performed to profile gene expression in OFT single-cells sorted from five *Wnt1^Cre^; Dullard^flox/+^; Rosa26^mTmG^* and five *Wnt1^Cre^; Dullard^flox/flox^; Rosa26^mTmG^* E11.5 embryos. At least 70 cells were isolated per gate and genotype (GFP⁺, CD31⁺, RFP⁺). (**B**) Graph showing the distribution of all cells analysed (dots) as a function of normalised expression values of the house keeping genes *Actb* and *Gapdh* and a linear regression (red line). (**C**) t-SNE plot showing the distribution of 44-genes-based transcriptomes of 433 OFT cells expressing the indicated markers isolated from both *Wnt1^Cre^; Dullard^+/flox^; Rosa26^mTmG^* and *Wnt1^Cre^; Dullard^flox/flox^; Rosa26^mTmG^* E11.5 embryos. (**D**) Unsupervised clustering heatmap of the 433 OFT isolated cells from *Wnt1^Cre^; Dullard^+/flox^; Rosa26^mTmG^* and *Wnt1^Cre^; Dullard^flox/flox^; Rosa26^mTmG^* based on the gene expression level of the 44 genes included in the panel (**Supplementary file 1**). Six different groups of cells can be discriminated, among which the endocardial (Group 1) cells expressing high levels of *Flt1*, *Kdr*, *Nfatc1* and *Tek*, and the epicardial (Group 5) cells expressing high levels of *Wt1*, *Tcf21*.

atrophy. While in other biological systems, Dullard loss has been associated with decreased Wnt-*β*-Catenin signalling (*Tanaka et al., 2013*) or enhanced TGF*β*-Smads2/3 signalling (*Hayata et al., 2015*), these signalling pathways are probably not affected upon Dullard deletion in the cardiac NCC. On the one hand, we showed that the phosphorylation state of Smad2 in Dullard mutant and control cardiac NCC was comparable (*Figure 1—figure supplement 1E*). On the other hand, loss of Wnt signalling in cardiac NCC leads to persistent truncus arteriosus, which is associated with reduced NCC-mediated cushion fusion (*Brade et al., 2006*). Furthermore, gain- and loss-of-function

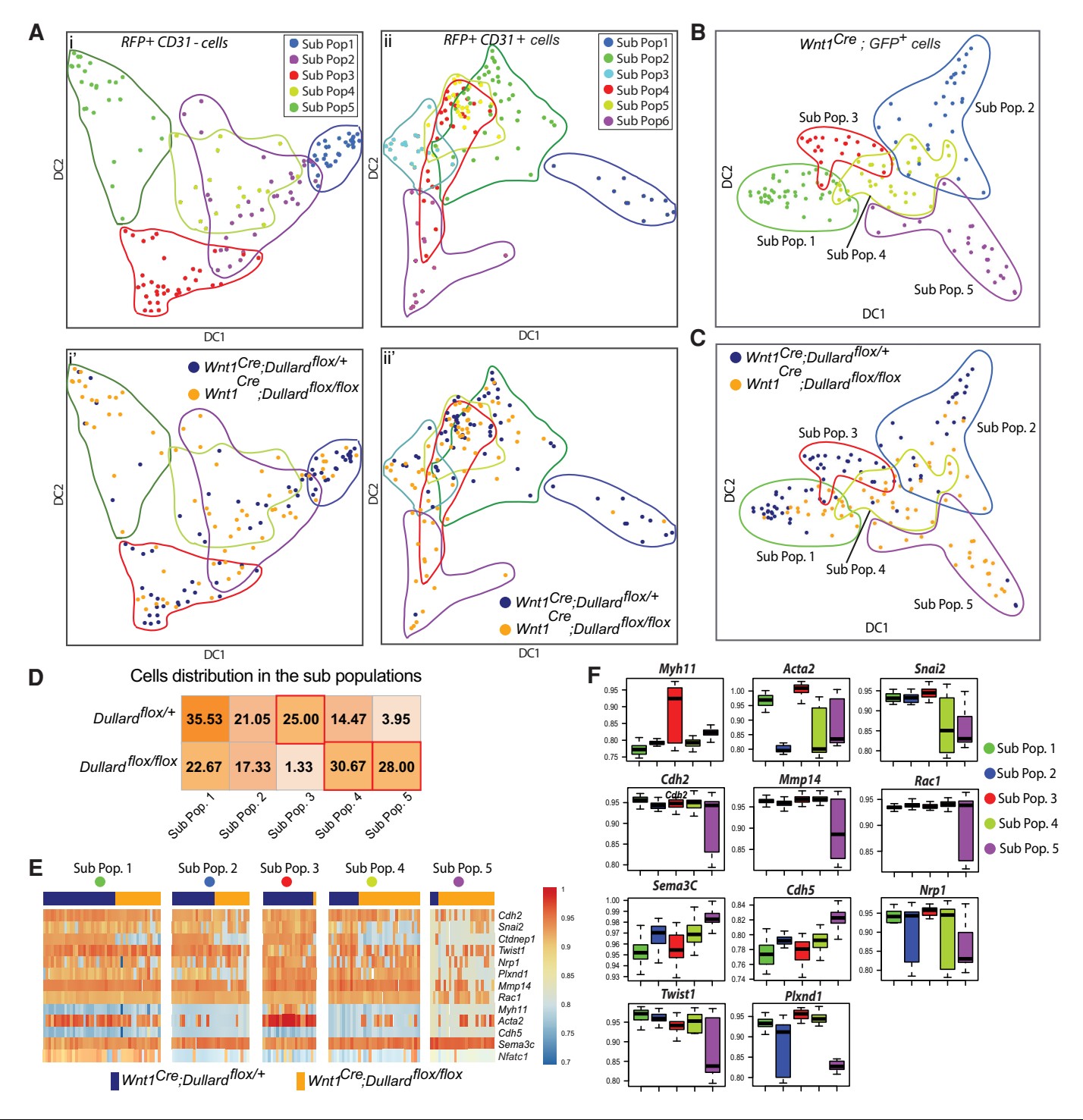

**Figure 5.** Impact of Dullard deletion in NCC on the transcriptomic variations within the distinct cellular subtypes of E11.5 hearts. (**A**) Projected position of 44 genes-based transcriptomes assessed in mutant and control CD31+; RFP+ and CD31-; RFP+ OFT cells on diffusion maps made using the first two Diffusion Component 1 (DC1) and 2 (DC2) (*Figure 5—figure supplement 1B*). Sub-populations defined with hierarchical clustering are presented in i and ii, while the genotype of cells is illustrated in i' and ii'. (**B, C**) Projected position of 44 genes-based transcriptomes assessed in mutant and control cardiac GFP+ NCC on diffusion maps made using the first two Diffusion Component 1 (DC1) and 2 (DC2) (see *Figure 5—figure supplement 2B*). The five subpopulations defined in *Figure 3A* are highlighted. (**D**) Percentage of cardiac NCC in each subpopulation. (**E**) Heatmaps showing the levels of expression of selected genes in all GFP+ NCC in the five subpopulations identified using unsupervised hierarchical clustering (*Figure 5—figure supplement 2A*) coming from control (blue) or mutant (orange) E11.5 embryos. (**F**) Boxplot representation of the expression levels of genes

*Figure 5 continued on next page*

*Figure 5 continued*

differentially expressed between the five NCC subpopulations (Sub-Pop 1: 44 cells, Sub-Pop 2: 29 cells, Sub-Pop3: 20 cells, Sub-Pop 4: 34 cells, Sub-Pop 5: 24 cells) (mean ±s.d.).

The online version of this article includes the following figure supplement(s) for figure 5:

**Figure supplement 1.** Transcriptional profiling of CD31+; RFP+ OFT cells.

**Figure supplement 2.** Single-cell transcriptional profiling of GFP⁺ NCC at E11.5.

experiments argue that BMP levels stand as a sufficient parameter to tune the degree of condensation of cardiac NCC towards the endocardium. Lowering BMP levels was shown to prevent fusion of the cardiac cushions and leads to persistent truncus arteriosus (*Jia et al., 2007*; *Stottmann et al., 2004*). Instead, we showed that increasing downstream BMP signalling due to Dullard loss triggers premature cardiac NCC condensation to the endocardium and accentuated OFT septation (*Figure 2*). This phenotype is reminiscent of the misplaced septation and narrowing of one of the OFT tubes observed in mouse mutants for the BMP and TGFβ signalling antagonist Smad6 (*Galvin et al., 2000*). As such, the BMP signalling gradient can be seen as a means that allow septation of the OFT to proceed in a zipper-like fashion. This is further supported by the fact that altering the ability of cardiac NCC to contract, migrate and adhere, hence to condense, prevents the correct formation and positioning of the aorticopulmonary septum (*Luo et al., 2006*; *Phillips et al., 2013*; *Plein et al., 2015*).

Thirdly, we have uncovered part of the cellular and molecular mechanism by which Dullard controls cardiac NCC behaviour. Dullard deletion triggers a downregulation of mesenchymal markers reminiscent of a cardiac NCC transition towards epithelial-like states, with a loss of migratory freedom and increased cohesiveness between cells (*Kim et al., 2017*). Furthermore, out of the transcriptional changes induced by the loss of *Dullard*, several lines of evidence support the idea that the upregulation of *Sema3c* is likely to play a predominant role in the increased condensation of the cardiac NCC. In fact, the expression of *Sema3c* in the cardiac NCC is required for their convergence to the endocardium and was also shown to promote the aggregation of cardiac NCC in primary cultures as well as in cancer cells in vivo (*Delloye-Bourgeois et al., 2017*; *Feiner et al., 2001*; *Kodo et al., 2017*; *Plein et al., 2015*; *Toyofuku et al., 2008*). In addition, the Sema3C-Nrp1 signalling has been proposed as a triggering signal for the EMT of the endocardium which accompanies its rupture (*Plein et al., 2015*). Intriguingly, our data show that the elevation of *Sema3c* in Dullard mutants is associated with variations in the expression of EMT factors in the endocardium (*Figure 5—figure supplement 1*), that would indicate that either the endocardium EMT is inhibited or has taken place prematurely. Testing the first possibility would require a fine quantification over time of rare EMT figures within the endocardium of control and Dullard mutants at several developmental stages. Yet, in the light of the premature septation of the endocardium we observed in Dullard mutants, we favour the second possibility. The elevation of *Sema3c* in Dullard mutants raises also the question about the nature of the regulatory relationship existing between BMP signalling and *Sema3c* expression. Embryos from the *Pax3^Cre* driver line harbour myogenic recombined cells which, as the cardiac NCC, show a striking increase in phosphorylation of Smad1/5/8 and *Sema3c* expression when *Dullard* is deleted (*Figure 1—figure supplement 1E*). This suggests that the regulatory influence of BMP signaling on *Sema3c* expression is not restricted to the context of cardiac NCC but also to other cell types. However, it remains unclear if Smads act directly on *Sema3c* expression or indirectly via an intermediate transcription factor. In fact, Gata6, a member of the zinc finger family of transcription factors, has been described as an activator of *Sema3c* expression in the cardiac NCC (*Kodo et al., 2017*; *Lepore et al., 2006*). Yet, we did not observe any significant difference of *Gata6* expression in *Dullard* deleted cardiac NCC (data not shown).

Finally, our data indicates that mice in which Dullard is deleted in the NCC harbours the traits of patients affected by Fallot's tetralogy condition, and thus represents one of the rare animal models for this pathology (*Neeb et al., 2013*). Interestingly, while the few well-characterized genetic causal factors identified in humans (such as Jag1, Nkx2.5, Tbx5) and the analysis of few mouse mutants would primarily incriminate defects at the level of the SHF, our data suggest that this sheet of cells is unaffected in Dullard mutants (*Neeb et al., 2013*; *Morgenthau and Frishman, 2018*). Instead, our

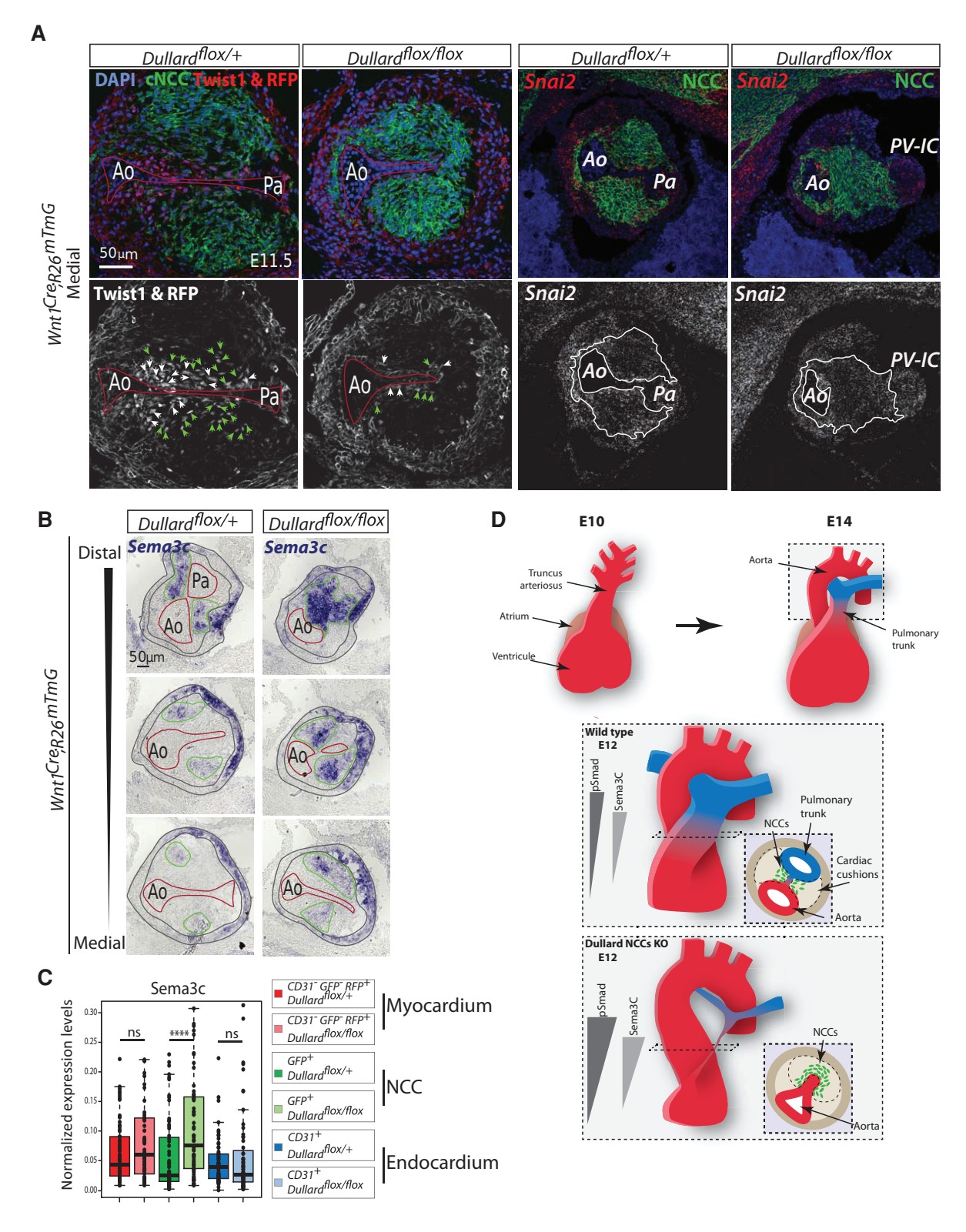

**Figure 6.** Dullard prevents NCC from acquiring epithelial-like traits and prolongs the expression of mesenchymal drivers. (E) Twist1 immunolabelling and RFP signal (red; grey), *Snai2* mRNA distribution assessed by RNAscope (red; grey), GFP (green) immunolabelling and DAPI staining on transverse sections through the medial OFT of E11.5 *Wnt1^Cre*; *Dullard^flox/+*; *Rosa26^mTmG* and *Wnt1^Cre*; *Dullard^flox/flox*; *Rosa26^mTmG* embryos. The red lines mark the endocardium, green arrowheads point at NCC, while the white ones indicate the endocardium. The white lines delineate the cardiac NCC cushions.
*Figure 6 continued on next page*

*Figure 6 continued*

RFP and GFP mark cell membrane whereas Twist1 is cytoplasmic or nuclear. (B) *Sema3c* expression assessed by ISH on transverse sections from distal to medial OFT levels of E11.5 *Wnt1^Cre; Dullard^flox/+* and *Wnt1^Cre; Dullard^flox/flox* embryos. The endocardium is delineated with a red line, the cardiac NCC areas with a green line and the myocardium with grey lines. (C) Normalized expression levels of *Sema3c* assayed by q-RT-PCR on single cells isolated after immuno-marking endothelial CD31^+ cells from E11.5 *Wnt1^Cre; Dullard^flox/+* and *Wnt1^Cre; Dullard^flox/flox; Rosa26^mTmG* hearts (dots: value for a single cell; boxplot: mean ± s.e.m.). (D) Model for the molecular and cellular cues controlling OFT septation. Upper panel: Morphogenesis of the single truncus arteriosus at E10.5 into fully formed great arteries at E14.0. Middle panel: Shape of P-Smad1/5/8 and Sema3c gradients along the OFT distal-proximal axis in a control situation. NCC in the cardiac cushions condense toward the endocardium between the aorta and the pulmonary trunk. Lower panel: Premature condensation of NCC at the pulmonary trunk in absence of Dullard in NCC, is associated with increased levels of Sema3c and BMP signalling.

The online version of this article includes the following figure supplement(s) for figure 6:

**Figure supplement 1.** Dullard loss in cardiac NCC alters the expression of *Sema3c* specifically in NCC.

data provide further evidence for the NCC as one of the OFT cell lineages whose development defects can lead to the formation of hearts with an aorta overriding the interventricular septum (*High et al., 2007*). Further explorations would be required to assess whether the malformations led by defects in the NCC or the SHF are comparable and whether, in both cases, the molecular and cellular defects lead to more pressure on the endocardium and forces its rupture.

# Materials and methods

## Key resources table

| Reagent type (species) or resource | Designation | Source or reference | Identifiers | Additional information |
|---|---|---|---|---|
| Strain, strain background (*Mus musculus*) | C57BL/6JRj | Janvier Labs | | |
| Genetic reagent (*Mus musculus*) | Wnt1^Cre | PMID: 9843687 | MGI:2386570 | Dr A Pierani (Imagine Institute) |
| Genetic reagent (*Mus musculus*) | Pax3^Cre | PMID: 15882581 | MGI: 3573783 | Dr F Relaix (Créteil University) |
| Genetic reagent (*Mus musculus*) | Rosa26^mTmG | PMID: 17868096 | MGI: 3716464 | Dr F Relaix (Créteil University) |
| Genetic reagent (*Mus musculus*) | Dullard^Flox | PMID: 23360989 | other | Dr R Nishinakamura (Kumamoto University) |
| Antibody | Anti-GFP (Chicken) | Aves Labs | GFP-1020 RRID:AB_10000240 | IF(1:500) |
| Antibody | anti-PECAM (Rat monoclonal) | Santa-Cruz Biotechnology | Sc-18916 RRID:AB_627028 | IF(1:200) |
| Antibody | Anti-phosphoSmad1/5/8 (Rabbit monoclonal) | Cell Signalling Technology | 13820S RRID:AB_2493181 | IF (1:500) |
| Antibody | Anti-Myosin Heavy Chain (mouse monoclonal) | DSHB | MF20 RRID:AB_2147781 | IF (1:300) |
| Antibody | Anti-phospho-histone H3 (rabbit polyclonal) | Cell Signalling Technology | 9701 RRID:AB_331535 | IF (1:500) |
| Antibody | Anti-cleaved Caspase 3 (Rabbit monoclonal) | Cell Signalling Technology | 9664 RRID:AB_2070042 | IF (1:500) |
| Antibody | Anti-Isl1 (Rabbit polyclonal) | Abcam | Ab20670 RRID:AB_881306 | IF (1:500) |
| Antibody | AF 488 donkey IgG anti-chicken IGG (H+L) | Interchim | 703-545-155 RRID:AB_2340375 | IF (1:500) |
| Antibody | Alexa Fluor 555 Goat Anti-Rabbit IgG (H+L), highly cross-adsorbed | Life Technologies | A-21429 RRID:AB_2535850 | IF (1:500) |

*Continued on next page*

*Continued*

| Reagent type (species) or resource | Designation | Source or reference | Identifiers | Additional information |
|---|---|---|---|---|
| Antibody | AF 647 Donkey IgG Anti Rabbit IGG (H+L) | Interchim | 711-605-152 RRID:AB_2492288 | IF (1:500) |
| Antibody | Alexa Fluor 647 Goat Anti-Rat IgG (H+L) | Life Technologies | A21247 RRID:AB_141778 | IF (1:500) |
| Antibody | Alexa Fluor 647 Goat Anti-Mouse IgG (H+L) Antibody, highly cross-adsorbed | Life Technologies | A-21236 RRID:AB_2535805 | IF (1:500) |
| Antibody | Alexa Fluor 488 Goat Anti-Mouse IgG (H+L) Antibody, highly cross-adsorbed | Life Technologies | A-11029 RRID:AB_2534088 | IF (1:500) |
| Antibody | Alexa Fluor 488 Goat Anti-Rabbit IgG (H+L) Antibody | Life Technologies | A-11008 RRID:AB_143165 | IF (1:500) |
| Antibody | Anti-Phospho Smad 2/3 (Rabbit monoclonal) | Ozyme | 8828 s RRID:AB_2631089 | IF (1:200) |
| Sequence-based reagent | Sema3C | PMID: 16397144 | other | RNA probe from Dr S Zaffran (Aix Marseille University) |
| Sequence-based reagent | Dullard | Advanced Cell Diagnostics | #456911 | RNA probe |
| Sequence-based reagent | Sema3C | Advanced Cell Diagnostics | #441441 | RNA probe |
| Sequence-based reagent | Twist1 | Advanced Cell Diagnostics | #414701 | RNA probe |
| Sequence-based reagent | Snai2 | Advanced Cell Diagnostics | #451191 | RNA probe |
| Sequence-based reagent | Msx2 | Advanced Cell Diagnostics | #421851 | RNA probe |
| Sequence-based reagent | Id2 | Advanced Cell Diagnostics | #445871 | RNA probe |
| Antibody | Anti-Ter119 Pe-Cy7 (Mouse monoclonal) | Sony | 1181110 | Facs (1:300) |
| Antibody | Anti-CD45 APC-Cy7 (Mouse monoclonal) | BD Pharmingen | 557659 RRID:AB_396774 | Facs (1:300) |
| Antibody | Anti-CD31 APC (Mouse monoclonal) | BD Pharmingen | Clone MEC 13.3 Catalog No. 561814 RRID:AB_10893351 | Facs (1:300) |
| Commercial assay or kit | 7AAD PE-Cy7 | BD Pharmingen | 559925 | Facs (1:800) |
| Commercial assay, kit | CellsDirect One-Step qRTPCR Kit | Invitrogen | 11753100 | |
| Commercial assay, kit | 48.48 Sample/Loading Kit— 10 IFCs | Fluidigm Corporation | BMK-M10- 48.48 | |
| Cell Line (*Mus musculus*) | C2C12 | ATCC, PMID: 28966089 | CRL-1772, RRID:CVCL_0188 | |
| Peptide, recombinant protein | Human BMP-2 | Thermo Fisher Scientific | #PHC7145 | 50 ng/ml |
| Software, algorithm | R, pHeatmap (v1.0.10) | R foundation | R package (v3.2.2), RRID:SCR_016418 | |
| Software, algorithm | R, phenograph (v0.99.1) | R foundation | R package (v3.2.2), RRID:SCR_016919 | |

*Continued on next page*

*Continued*

| Reagent type (species) or resource | Designation | Source or reference | Identifiers | Additional information |
|---|---|---|---|---|
| Software, algorithm | R, ggplot2 (v3.1.0) | R foundation | R package (v3.2.2), RRID:SCR_014601 | |
| Software, algorithm | R, Destiny (v2.6.1) | R foundation | R package (v3.6) | https://github.com/ theislab/destiny |
| Recombinant DNA reagent | pEGFP-Dullard (plasmid) | This paper | | GFP-Dullard expression plasmid |
| Recombinant DNA reagent | pEGFP-Dullard D67E (plasmid) | This paper | | GFP-Dullard D67E expression plasmid |
| Recombinant DNA reagent | pEGFP-N1 | Clontech | 6085–1 | |
| Commercial assay or kit | Gateway LR Clonase | ThermoFisher | 11791100 | |
| Commercial assay or kit | Taq Polymerase, Superscript III | Thermofisher | 11732020 | |
| Sequence-based reagent | Dull_FL_C1_FW | This paper | PCR primers | GGGGACAAGTTTGTAC AAAAAAGCAGGCTTAAT GATGCGGACGCAGTGT |
| Sequence-based reagent | Dull_FL_C1_Rev | This paper | PCR primers | GGGGACCACTTTGTAC AAGAAAGTGGGTCTCAC CAGAGCCTATGTTGGTG |
| Sequence-based reagent | Dull_D67E_FW | This paper | PCR primers | GATCCTGGTGCTGGA ACTGGACGAAACCCTG |
| Sequence-based reagent | Dull_D67E_Rev | This paper | PCR primers | CAGGGTTTCGTCCAG TTCCAGCACCAGGATC |

## Animals

All animal experiments were approved by the Animal Ethics Committee of Sorbonne University. We used the mouse strains described in the following papers and MGI IDs: *Dullard*$^{flox/flox}$ (*Sakaguchi et al., 2013*; in these mice exons 2 to 4 are floxed), *Pax3*$^{Cre}$ (*Engleka et al., 2005*, MGI: 3573783), *Wnt1*$^{Cre}$ (*Danielian et al., 1998*, MGI:2386570), *Rosa26*$^{mTmG}$ (*Muzumdar et al., 2007*, MGI: 3716464), and *C57BL/6JRj* (Janvier Labs).

## Immunohistochemistry and imaging

Mouse embryos were collected at E11.5 and dissected in cold PBS, incubated 5 min in 200 mM KCl to stop heart beating and fixed for 2–3 hr in 4% PFA (Electron Microscopy Science, #15710S) at 4°C.

For immunostaining on cryosections, embryos were cryoprotected in 20% sucrose overnight at 4°C, embedded in OCT, and cryosectioned at 12 µm thickness. Sections were permeabilized 10 min in PBS/0.5% Triton, incubated for 1 hr in blocking buffer (5% goat serum in PBS) and overnight in primary antibody solution (in 1% BSA in PBS). After thorough washing in PBS, they were incubated 1 hr in secondary antibody solution (in 1% BSA in PBS), washed in PBS and mounted in Fluoromount-G (Clinisciences, 0100–01). Immunostainings were acquired using a Nikon Ti2 microscope, driven by Metamorph (Molecular Devices), equipped with a motorized stage and a Yokogawa CSU-W1 spinning disk head coupled with a Prime 95 sCMOS camera (Photometrics), then assembled and analyzed on Fiji (*Schindelin et al., 2012*).

For wholemount staining, we followed the 3Disco protocol (*Belle et al., 2017*) to immunostain, clear and image with a lightsheet ultramicroscope (LaVision BioTec). Alternatively, micro-dissected hearts immunostained as described in *Belle et al. (2017)*, were clarified with BABB and imaged using a LSM700 confocal microscope (Carl Zeiss) (*Gopalakrishnan et al., 2015*). 3D renderings were generated using the Imaris software.

The primary antibodies used were raised against: GFP (chicken, Aves Labs, GFP-1020, 1/500), Pecam (rat monoclonal, Santa-Cruz Biotechnology, sc-18916, 1/200), Phospho-Smad1/5/8 (rabbit monoclonal, Cell Signalling Technology, 13820S, 1/500), myosin heavy chain, MyHC (mouse monoclonal, DSHB, MF20, 1/300), Phospho-Histone H3 (rabbit, Cell Signalling Technology, 9701, 1/500),

Cleaved Caspase-3 (rabbit monoclonal, Cell Signalling Technology, Asp175, 1/500), Isl-1 (Abcam, ab20670), Phospho-Smad2,3 (rabbit, Ozyme, 8828 s, 1:200). Secondary antibodies were bought from Life technologies or Interchim and were Donkey or Goat Igg coupled to Alexa fluorophores.

## In situ hybridisation

The *Sema3c* probe was provided by the lab of S. Zaffran (Bajolle et al., 2006). In situ hybridization on cryosections were processed following the protocol described in Chotteau-Lelièvre et al. (2006).Fluorescent in situ hybridization probes for *Dullard* (#456911), *Sema3c* (#441441), *Twist1* (#414701), *Snai2* (#451191), *Msx2* (#421851) and *Id2* (#445871) were obtained from Advanced Cell Diagnostics, Inc In situ hybridization was performed using the RNAscope V2-fluorescent kit according to the manufacturer's instructions. For sample pre-treatments: H2O2 treatment was performed during 10 min at RT, retrieval 2 min at 98°C and slides were digested with Protease Plus reagent for 15 min at 40°C. After the probe detection steps immunostaining was performed as described above with fluorescent secondary antibodies. Sections were imaged using a 40x objective on a LSM700 microscope (Zeiss) or Nikon Ti2 microscope.

## Image analyses, quantification, statistical analysis

Mean levels of P-Smad1/5/8 in the cardiac NCC were quantified using Image J thanks to a mask established on the GFP channel. Distances between cardiac NCC and between cardiac NCC and the endocardium, as well as the angle of the major axis of NCC to an axis linking the Ao and Pa poles of the endocardium were measured using Metamorph (Molecular Devices) and a home-made algorithm on Excel. Statistical analysis was performed with the Student's t-test or Mann-Whitney test depending on normality. The analysis was performed using Prism Software (GraphPad). Statistical significance is represented as follows: ***$p < 0.001$. All results are shown as mean ± standard deviation.

## Plasmids

Dullard was directly cloned from an NIH 3T3 mRNA library using the SuperScript III One-Step RT-PCR System (Life Technologies) using primers Dull_FL_C1_FW 5´-GGGGACAAGTTTG TACAAAAAAGCAGGCTTAATGATGCGGACGCAGTGT-3' and Dull_FL_C1_Rev 5´-GGGGACCAC TTTGTACAAGAAAGTGGGTCTCACCAGAGCCTATGTTGGTG-3' for N-terminal tag destination vectors (pDONR221). For phosphatase-null point mutant Dullard D67E, site-directed mutagenesis was performed by PCR amplification of pDONR 221 Dullard full-length vector using primers Dull_-D67E_FW 5´-GATCCTGGTGCTGGAACTGGACGAAACCCTG-3' and Dull_D67E_Rev 5´-CAGGG TTTCGTCCAGTTCCAGCACCAGGATC-3' followed by DpnI endonuclease mediated digestion of the parent (methylated) DNA chain. After sequence confirmation, entry vectors were recombined with pEGFP GW C1 for N-terminal (Life Technologies) GFP-tag fusion proteins using the Gateway system (Thermofisher).

## Cell culture and transfection

C2C12 cells were cultivated at 37°C/5% CO2, in growth medium (DMEM, 4.5 g/L, D-glucose, 4 mM L-glutamine, 1 mM sodium pyruvate, 10% fetal calf serum). Plasmid transfection was performed using Lipofectamine 2000 (Life Technologies). 24 hr after transfection, BMP2 recombinant human protein (Thermo Fisher Scientific, #PHC7145) was applied for 1 hr on cells. Cells were fixed with 4% PFA and immunostained with the phospho-Smad1/5/8 antibody. Alternatively, cells were washed in PBS, collected with PBS 1% SDS and passed through Qiashredder columns (Qiagen) to disrupt nucleic acids. Proteins extracts were then processed for western blotting using pre-cast gels (Life Technologies) and transferred on nitrocellulose by semi-dry transfer (Bio-Rad). C2C12 (ATCC CRL-1772) is a murine myogenic cell line. Its myogenic profile is regularly checked by placing them in low serum to trigger the formation of myotubes through cell-cell fusion. Their mycoplasma contamination status resulted negative.

## Tissue dissociation and FACS sorting

Mouse embryos were collected at E11.5 and placed in HBSS/1% FBS (HBSS +/+, Invitrogen) during genotyping. OFT were micro-dissected and dissociated by 15 min incubation in collagenase (0.1 mg/ml in HBSS, C2139 Sigma) and thorough pipetting. HBSS (10% FBS) was added to the cells

medium to stop the enzymatic reaction. OFT cell suspensions were centrifuged and resuspended in HBSS (1% FBS) before immunostaining. The panel of conjugated antibodies used for FACS included Ter119 Pe-Cy7 (Erythroid Cells, anti-mouse, Sony, Catalog No. 1181110, 1/300), CD45 APC-Cy7 (Rat anti-mouse, BD Pharmingen, Catalog No. 557659, 1/300), CD31 APC (Rat anti-mouse, BD Pharmingen, Clone MEC 13.3 Catalog No. 561814, 1/300) diluted in HBSS (1% FBS). Cells were centrifuged and resuspended in the antibody solution for a 25 min incubation period (4°C, dark), washed three times, filtered (Fisher cell strainer, 70 µm mesh) and 7AAD PE-Cy7 (1/800) was added in the cells suspension to exclude dead cells. Cells were sorted in a BD FACSAria III into 96-well plates loaded with RT-STA reaction mix (CellsDirect One-Step qRTPCR Kit, Invitrogen) and 0.2x specific TaqMan Assay mix (see *Supplementary file 1* for assays list).

## Single-cell gene expression

We proceeded as described in *Valente et al. (2019)*. Cells were sorted in RT-STA reaction mix from the CellsDirect One-Step qRT-PCR Kit (Life Technologies), reverse transcribed and specific target pre-amplified (20 cycles), according to the manufacturer's procedures. Pre-amplified samples were diluted 5x with low EDTA TE buffer prior to qPCR analysis using 48.48 Dynamic Array IFCs and the BioMark TM HD System (Fluidigm). The same TaqMan gene expression assays (20x, Life Technologies) were individually diluted 1:1 with 2x assay loading reagent (Fluidigm). Pre-amplified samples were combined with TaqMan Universal Master Mix (Life Technologies) and 20x GE sample loading reagent (Fluidigm). Loading of the 48.48 Dynamic Array TM IFCs and qPCR cycling conditions followed the Fluidigm procedure for TaqMan gene expression assays.

## Bioinformatic analysis

The analytic framework used followed the one described in *Perchet et al. (2018)*; *Valente et al. (2019)*. It included notably a normalization of all the cycle threshold (Ct) values extracted from the Biomark chips using the mean value of *Actb* and *Gapdh* housekeeping genes (*Figure 4B*). For visualisation of the single-cell multiplex qPCR, done on 44 genes, we generated a heatmap using the pHeatmap (v1.0.10) R package (v3.2.2). For unsupervised clustering, we used PhenoGraph that takes as input a matrix of N single-cell measurements and partitions them into subpopulations by clustering a graph that represents their phenotypic similarity. PhenoGraph builds this graph in two steps. Firstly, it finds the k nearest neighbors for each cell (using Euclidean distance), resulting in N sets of k-neighborhoods. Secondly, it operates on these sets to build a weighted graph such that the weight between nodes scales with the number of neighbors they share. The Louvain community detection method is then used to find a partition of the graph that maximizes modularity. Given a dataset of N d-dimensional vectors, M distinct classes, and a vector providing the class labels for the first L samples, the PhenoGraph classifier assigns labels to the remaining N_L unlabeled vectors. Firstly, a graph is constructed as described above. The classification problem then corresponds to the probability that a random walk originating at unlabeled node x will first reach a labeled node from each of the M classes. This defines an M-dimensional probability distribution for each node x that records its affinity for each class. PhenoGraph is implemented as Rphenograph (v0.99.1) R package (v3.2.3).

We used boxplot for gene expressions of clusters obtained with PhenoGraph algorithm, from package ggplot2 (v3.1.0) R package (v3.2.2). For visualisation and pathway analyses (tree), Destiny (v2.6.1) on R package (v3.6) generates the diffusion maps. Destiny calculates cell-to-cell transition probabilities based on a Gaussian kernel with a width σ to create a sparse transition probability matrix M. For σ, Destiny employs an estimation heuristic to derive this parameter. Destiny allows for the visualization of hundreds of thousands of cells by only using distances to the k nearest neighbors of each cell for the estimation of M. An eigen decomposition is performed on M after density normalisation, considering only transition probabilities between different cells. The resulting data-structure contains the eigenvectors with decreasing eigenvalues as numbered diffusion components (DC), the input parameters and a reference to the data. These DC are pseudotimes identifying differentiation dynamics from our sc-qPCR data.

Impact of gene expressions creating the different dimensions is represented as horizontal box plots, showing cells up- or down-regulating indicated genes in the plot.

## Ultrasound imaging

Pregnant mice were anaesthetised (3% isoflurane in air and maintained at 1.5%), installed on a heating pad and monitored for respiration frequency, ECG and temperature (Vevo Imaging Station, Visualsonics). Pregnant mice were intraperitoneally injected with Metacam (1mg/kg body weight; Boehringer Ingelheim). A laparotomy was then performed and the uterine horns were gently exteriorized to allow direct visualisation of embryos using a high-resolution ultrasound imaging scanner (VEVO2100, Visualsonics) equipped with a 60 MHz probe (MS-700). To ensure contact between the ultrasound probe and the embryos, a warm sterile gel (Aquasonic) was used. Long axis view of the heart was performed on each embryo to measure left and right ventricle dimensions. PW Doppler measurements were achieved positioning the caliper on the descending abdominal aorta with embryo in sagittal view. From Abdominal Aorta the following measurements were performed: Abdominal Aorta Peak Systolic Velocity (AA PSV, mm/s), End Diastolic Velocity (AA EDV, mm/s), Velocity Time Integral of mean velocities (AA VTI, ms). From these measurements, the following parameters were calculated: Mean Velocity (AA VTI Mean Vel), Abdominal Aorta Pulsatility Index (AA PI), Abdominal Aorta Resistive Index (AA RI).

## Acknowledgements

We thank the Cadot, Bitoun and Ribes Laboratories for discussions, Sigolène Meilhac for her help on understanding heart development concepts, Edgar Gomes laboratories, Isabelle Le Roux and Pascale Gilardi-Hebenstreit for useful comments, Stéphane Zaffran for the *Sema3c* ISH probe and Morgane Belle for Lightsheet microscopy. The *Pax3^Cre* and *Rosa26^mTmG* transgenic lines were kindly provided by F Relaix, and the *Wnt1^Cre* line by A Pierani. This work was supported by Agence Nationale pour la Recherche (ANR-14-CE09-0006-04) to BC; Association Institut de Myologie to BC. VR is an INSERM researcher, and work in her lab is supported by CNRS/INSERM ATIP-AVENIR program, as well as by a Ligue Nationale Contre le Cancer grant (PREAC2016.LCC). MV has a postdoctoral fellowship from the Laboratoire d'Excellence Revive (Investissement d'Avenir; ANR-10-LABX-73). In vivo imaging was performed at the Life Imaging Facility of Paris Descartes University (Plateforme Imageries du Vivant – PIV) and is supported by France Life Imaging (grant ANR-11-INBS-0006).

## Additional information

### Funding

| Funder | Grant reference number | Author |
|---|---|---|
| Agence Nationale de la Recherche | ANR-14-CE09-0006-04 | Bruno Cadot |
| Ligue Contre le Cancer | PREAC2016.LCC | Vanessa Ribes |
| Agence Nationale de la Recherche | ANR-10-LABX-73 | Mariana Valente |

The funders had no role in study design, data collection and interpretation, or the decision to submit the work for publication.

### Author contributions

Jean-François Darrigrand, Conceptualization, Data curation, Formal analysis, Investigation, Visualization, Methodology; Mariana Valente, Data curation, Investigation, Visualization, Methodology; Glenda Comai, Data curation, Formal analysis, Validation, Investigation, Visualization; Pauline Martinez, Gilles Renault, Carmen Marchiol, Investigation; Maxime Petit, Formal analysis, Validation, Investigation; Ryuichi Nishinakamura, Daniel S Osorio, Resources; Vanessa Ribes, Conceptualization, Data curation, Supervision, Validation, Investigation, Visualization, Methodology, Project administration; Bruno Cadot, Conceptualization, Resources, Data curation, Supervision, Funding acquisition, Validation, Investigation, Visualization, Methodology, Project administration

## Author ORCIDs

Jean-François Darrigrand [iD] https://orcid.org/0000-0002-5624-7585
Mariana Valente [iD] https://orcid.org/0000-0002-0735-6814
Glenda Comai [iD] https://orcid.org/0000-0003-3244-3378
Maxime Petit [iD] http://orcid.org/0000-0002-8443-1531
Daniel S Osorio [iD] https://orcid.org/0000-0003-4144-8189
Gilles Renault [iD] https://orcid.org/0000-0003-2273-1229
Vanessa Ribes [iD] https://orcid.org/0000-0001-7016-9192
Bruno Cadot [iD] https://orcid.org/0000-0002-1888-3898

## Ethics

Animal experimentation: All animal experiments (APAFIS#4163-2016042809186990) were approved by the Animal Ethics Committee of Sorbonne University (Permit Number: A751320).

## Decision letter and Author response

Decision letter https://doi.org/10.7554/eLife.50325.sa1
Author response https://doi.org/10.7554/eLife.50325.sa2

## Additional files

### Supplementary files

• Supplementary file 1. List of TaqMan gene expression assays (20x, Life Technologies) used for single-cell qPCRs experiments.

• Transparent reporting form

### Data availability

All data generated or analysed during this study are included in the manuscript and supporting files.

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
