## [Decision Letter]

**Acceptance summary:**

This paper, now much improved after key revisions, provides a compelling picture of the role of the phosphatase Dullard, a BMP signaling inhibitor, in patterning of the cardiac outflow tract, and specifically the contribution from cardiac neural crest cells. BMP signaling is established in a distal-high gradient along the outflow tract, and it is the magnitude of the gradient, not the gradient itself, that is moderated by Dullard. In the absence of Dullard, there is premature condensation of the cardiac neural crest in the outflow tract and breakdown of the endocardium, forcing a premature scission of the outflow into its two major arterial trunks, and in addition the pulmonary artery becomes constricted. The phenotype in rare surviving Dullard mutants closely resembles the tetralogy of Fallot, suggesting relevance to understanding of this severe congenital heart disease. The reviewers appreciated the clever use of single cell transcriptomics, which has helped to show which outflow cell types are affected and suggest the spatial elements of this dysfunction. The outflow tract is an important structure in heart development and evolution, and its complex makeup presents certain vulnerabilities during fetal life. This is an excellent study on how Dullard constrains BMP signaling to allow the correct extent and timing of tissue patterning in the outflow tract.

**Decision letter after peer review:**

Thank you for sending your article entitled "Transcriptional control of cardiac neural crest cells condensation and OFT septation by the Smad1,5,8 inhibitor Dullard" for peer review at *eLife*. Your article is being evaluated by three peer reviewers, one of whom is a member of our Board of Reviewing Editors, and the evaluation is being overseen by Marianne Bronner as the Senior Editor.

The reviewers have discussed the reviews with one another and the Reviewing Editor has drafted this decision to help you prepare a revised submission.

Summary:

The paper by Darrigrand et al. describes an analysis of mice mutant for the phosphatase gene Dullard in cardiac neural crest using both *Wnt1* and *Pax3*-driven (non-inducible) Cre drivers. The authors study morphology and gene/protein expression on sections with quantifications and study a 44-gene single cell transcriptome with clustering of cell types/states to explore changes in mutant populations. The authors describe a distal-proximal gradient of pSmad1/5/8 staining across the OFT and reduction in the level of staining but not the gradient, in Dullard conditional mutants. The most obvious morphological phenotype is a premature "rupture" of the endocardium and fusion of the endocardial cushions at distal and medical locations, in the absence of changes in NCC numbers or overall pattern or distribution. Morphological parameters of this phenotype, including cell proximity and orientation are described. Clustering analysis defines a number of GFP+ population states in wild type and mutant OFT, with skewing towards either wild type or mutant genotypes within predominantly two of these. Subsequent analysis identified up-regulation of *SemaC3* in NCCs as a possible driver of the premature rupture phenotype. The analysis of OFT defect has significant relevance to congenital defects of the OFT and great vessels, which are common. The work is interesting and relevant to understanding of heart development and disease, and the quantitative and single cell approaches are a strength. However, the reviewers have identified a number of important issues that need to be addressed before this work is ready for publication, as well as many other points for clarification. Major issues concern 1) the lack of detail on the temporal development and symmetry/asymmetry of NCC deployment into the OFT and how the phenotype evolves morphologically and in the context of related events such as EndMT; 2) the specificity of the phenotype with respect to BMP signaling; and 3) the possible role of Sema3C and other differentially-expressed genes in the development of the phenotype.

Essential revisions:

1) While Dullard is an established negative regulator of BMP/Smad1,5,8 activity, it is also a negative regulator of Tgfβ Smad2,3 phosphorylation and stability (Hayata et al., 2015). The current paper addresses the gradient in BMP/Smad1,5,8 activity in the OFT and establishes correlations with altered BMP activity and morphology but ignores the possibility that Tgfβ signalling or indeed other signaling pathways that may utilise this phosphatase may be involved. Experiments that address this: e.g. rescue of the phenotype with genetic or pharmacological dampening of BMP signalling will significantly strengthen the paper.

2) The model relies on gradients of BMP signalling activity across the OFT (high distal; low proximal). Figure 1E and Supplmentary Figure 1B show significant differences between control and Dullard mutants but not with respect to the proximal distal axis. Thus, this gradient cannot be claimed unless further work allows statistical significance to be established. Stats are also not shown for the distal-proximal change in OFT area in Supplementary Figure 2D.

3) Figure 2A left panels shows *Wnt1^Cre^* reporter expression in Pecam1^+^ OFT endothelial cells. How does this affect the results of the study, particularly the single cell gene expression analysis e.g. EMT markers? Does Sema3c expression in the OFT wall change in conditional mutant embryos? It appears to be reduced in the *Pax3*^Cre^ conditional Dullard mutant (Figure 6—figure supplement 1). The authors show that Dullard, like Sema3c, is also expressed in OFT myocardium. Can the authors add any conditional data addressing Dullard function in the OFT wall?

4) Subsection “Dullard is a key modulator of cardiac NCC mediated OFT septation” and Figure 2A. The authors claim that the distal and medial rupture of the OFT endocardium and imposition of condensed NCCs in Dullard mutants is somehow asymmetric, restricting the pulmonary artery lumen. The description is vague, and the claim is not convincing as in the controls also the Pa is considerably smaller than the aorta. The relevance of the comments about the PV-IC are not clear in this light and may not be justified expect that morphogenesis in mutants is abnormal. Overall, the study lacks a detailed temporal description of the development of NSC deployment and of the phenotype. Not having data at earlier stages of development (e.g. E9.5 and E10.5) leaves us guessing whether the observed phenotype (morphological, cellular, transcriptional) is the consequence of earlier problems during the ontogeny of NCCs. Not having later, fetal stages (e.g. E18.5) leaves us wondering what the morphogenetic consequences of dullard deletion are on the developed heart. For example, it would be important to establish whether the mutation leads to specific heart defects seen in congenital heart disease patients. The authors should provide information on the intercalated cushions. Do they form normally? Also, are there different behaviors in neural crest derived cells between the two main OFT cushions (this seems to be the case in the mutant cushions in Figure 2Avi'). Can the authors further investigate this with analysis of additional markers of OFT development such as Pitx2? The authors should discuss whether enriched Sema3c in the subpulmonary but not subaortic wall could be involved in asymmetric septation on upregulation of Sema3c in crest subsequent to Dullard loss. Finally, the significance of this asymmetric defect for congenital heart defects such as pulmonary trunk atresia and tetralogy of Fallot should be discussed.

5) Subsection “Dullard is a key modulator of cardiac NCC mediated OFT septation”. The mutant NCC more convincingly condenses prematurely in the medical region of the OFT. How does this relate to the model?

6) The OFT contains the junction between SHF-derived myocardium and NCC-derived smooth muscle cells. This point is ignored in this study and may have significance. It is noteworthy the Sema3c is expressed mostly in the presumed myocardium shown in medial and proximal sections (Figure 3F; Figure 6—figure supplement 1A) compared to distal sections (perhaps containing more NCC-derived SMCs). The authors mention other (myogenic) aspects of the *Pax3* conditional mutant phenotype – it seems relevant to expand on this as Sema3c appears also be upregulated. Do changes in Sema3C lead to defects in cell behavior?

7) Subsection “*Dullard* cell-autonomously controls cardiac NCC mesenchymal transcriptional state”. It can be presumed that Dullard works cell-autonomously, although the last sentence suggests that this was confirmed by the transcriptome data. How?

8) The attempts at spatial reconstruction of cell populations based on transcriptome data is admirable, although the story around Sema3c is simplistic and incomplete. Sema3c is regarded as a guidance See for example Kodo et al., 2017, and Plein et al., JCI 2015 for descriptions of interactions between NCC and SHF cells, and NCC and OFT endothelial cells, respectively. The role of EndMT and how it is affected in mutants is not addressed in this paper. The endothelial-proximal expression of Twist and Snail may relate to EndMT not condensation of NCC. The paper would be strengthened if this general aspect could be strengthened.

9) The single cell RT-PCR analysis is an informative approach that gives clear molecular insights into both the heterogeneity of OFT cell types and Dullard function. However, the analysis could be more complete. In particular the authors should provide more information on the different subpopulations of cells they define, including details of cell numbers for each subgroup. Group 3 appears to contain both RFP and GFP positive cells and expresses many genes expressed in mesodermal cardiac progenitor cells. This should be resolved further: is the grouping meaningful if it contains both GFP and RFP+ cells? Also, is *Tbx1*, for example, expressed in cardiac neural crest cells? Can the authors show this by immunofluorescence or other means? What fraction of all cardiac neural crest cells are positive for these mesodermal progenitor genes? When the authors refer to the muscle lineage in paragraph three of subsection “Dullard cell-autonomously controls cardiac NCC mesenchymal transcriptional state” presumably they mean smooth muscle, this should be specified.

10) Can the authors use their single cell data to identify any presumably indirect changes in endocardial gene expression that may mediate the endocardial rupture/fusion defect? Are the changes in Cdh5 expression in the crest lineage?

11) The authors should discuss whether the ingression of crest cells into the OFT may itself play a role in generating the BMP signaling gradient.

---

## [Author Response]

We thank the reviewers for their thoughtful and constructive comments. We have divided our response into three sections, the points highlighted by the reviewers and the editorial board. The answers to the specific points have been incorporated in these three points. We provide several new supplementary figures and the text has been modified accordingly.

A) Response to the general comments:

Point 1. The lack of detail on the temporal development and symmetry/asymmetry of NCC deployment into the OFT and how the phenotype evolves morphologically and in the context of related events such as EndMT.Point 1.1 A phenotypic characterization of Dullard mutant hearts anchored in a larger developmental window.

We now provide substantial phenotypic analysis of Dullard mutants throughout a larger time window than presented in the previous version of the manuscript, which bring insights into the role and time of action of the phosphatase during heart morphogenesis (Figure 2C-F—figure supplement 2).

First, labelling the cardiac NCC, the myocardium and/or the endocardium of E10.5 embryos indicates that the morphology of the OFT is not impacted at this stage by the loss of Dullard in the NCC compartment (Figure 2—figure supplement 1A). Hence, the premature NCC condensation in Dullard mutant OFT is likely established after E10.5.

Second, we now show that Dullard deletion using either the *Wnt1^cre^* or the *Pax3^cre^*drastically compromises the survival of embryos after E12.5 onwards (Figure 2C). Histological characterization of the rare embryos that can survive up to E14.5 or E18.5 revealed that their heart exhibits i) an opened interventricular septum, ii) a hypertrophy of the right ventricle, iii) an aorta overriding the two ventricles and finally a pulmonary stenosis (Figure 2D, E). Hence, as proposed by one of the reviewers, the phenotype of the heart developed in the presence of cardiac NCC missing Dullard is highly reminiscent to the heart of patients affected by the most cyanotic congenital heart disease, named tetralogy of Fallot (TOF) (Neeb et al., 2013). We now discuss how our work brings insight into the etiology of this condition, which is still poorly understood (cf. Snider and Conway, 2011).

Finally, we imaged the blood flow within the abdominal artery of E11.75 control and Dullard mutant embryos using doppler ultrasound (Figure 2F) (Nomura-Kitabayashi et al., 2009). It indicates that the flow is generally weaker in Dullard mutants compared to controls, as indicated by a lower systolic velocity peak in mutants than control. Hence the OFT defects triggered by the loss of the phosphatase in the NCC worsen the blood circulation of embryos and could in turn impair the survival rate of embryos.

Point 1.2 A better characterization of the effect of Dullard loss on the OFT cell types communicating with the NCC (experimental procedures detailed here will also address the questions raised in point 3).

We provide in our revised manuscript a better description of the differentiation and morphogenesis of the non-NCC derived OFT tissues. Overall these analyses sustain the idea that the NCC derived cardiac cushion is the tissue mostly affected by the loss of Dullard. No identity defect in the specification of the myocardium was found. Similarly, we had barely found any changes in the endocardium of Dullard mutants compared to controls. To illustrate this, we have:

– Performed bioinformatics analyses of our 44 gene base-transcriptomes centred on the RFP^+^;CD31^+^ endocardial cells and to the RFP^+^;CD31^-^ OFT cells following the pipeline used to describe GFP^+^ NCC transcriptomes (Figure 5A, Figure 5—figure supplement 1). As established for the GFP^+^ NCC, the endocardium cells and the RFP^+^;CD31^-^ OFT cells display transcriptomic variations and can be segregated into distinct populations (Figure 5Ai, ii). However, in contrast to the NCC cells, most of these populations all contain both control and mutant cells, with the exception of two populations of endocardial cells (Figure 5Ai’, ii’). The RFP^+^;CD31^+^ pop 1 is enriched for cells coming from control embryos, while the RFP^+^;CD31^+^ pop 6 contains more mutant cells than control cells. Again, in contrast to NCC case, very few of the genes that drive the segregation of the CD31^+^ cells into 6 populations display clear variations in the pop 6 or the pop 1 compared to the other pops (Figure 5—figure supplement 1B, C). *Twist1* stands as one of the pop1 signature genes and appears downregulated in the absence of Dullard. Accordingly, immune detection of this transcription factor revealed few positive Twist1^+^ endocardial cells in E11.5 control embryos, which were barely detectable I n mutant endocardium (Figure 6B).

– Brought further evidence that the *Wnt1^Cre^*and *Pax3^Cre^* drivers activity are not detectable in the endocardium to the reviewers. The yellow staining detected on the sections stained for GFP and PECAM correspond to overlapping membranes at sites of cell-cell interaction captured with a large pinhole confocal acquisition. FACS plots further showed a clear segregation between the GFP^+^ and the CD31^+^ cells (Author response image 1). Finally, within the NCC transcriptomes, we did not detect a cohort of genes which define endocardial cells (e.g. Figure 5—figure supplement 2). We have not included the data of the Author response images in the current version of the manuscript, as many papers have already extensively described the activity of these Cre drivers in the developing heart (Brown et al., 2001; Jiang et al., 2000).

– Performed several immunostaining or ISH for key markers of the myocardium, showing in agreement with transcriptomic data that the differentiation tissue and its pattern along thee bis not majorly affected by Dullard loss. Isl1 or myosin heavy chain (MF20) in E11.5 control and Dullard hearts displayed similar profile and expression levels in control and Dullard mutant embryos (Figure 2—figure supplement 1). Similarly, *Tbx1* and *Pitx2* expression patterns at the base of the pulmonary trunk and its expression level were comparable in control and mutant embryos (Author response image 2). Finally, more data on Sema3c mRNA distribution in Dullard mutants also indicate that Sema3c expression in the myocardial cuff is not altered by Dullard’s deletion in the NCC (Figure 1—figure supplement 1D).

N.B. We have tested several antibodies against the Cdh5, none of those were efficient enough to mark whether the elevated Cdh5 mRNA levels in Dullard mutant could also be detected at the protein level.

**Author response image 1. respfig1:** FACS plot showing that CD31+ cells are segregating from GFP+ NCC cells and the gating (coloured squares) used to collect cells for the q-RT-PCR screening.

**Author response image 2. respfig2:** RNAscope for *Tbx1* and Pitx2 showing there are no differences between Control and mutants PFT for their expression levels.

Point 2. The specificity of the phenotype with respect to BMP signalling.

We now provide further support for the lack of Dullard leading to an elevation of BMP signalling.

– A new statistical test (two way-Anova) indicate that the variations in PSmads levels both between control and mutant cells at a given proximo-distal axis level and between the distal and proximal regions of the OFT in a given genetic background are significant (significance for the differences between *Wnt1^cre^,Dullard^Flox/Flox^* and *Wnt1^cre^;Dullard^Flox/+^*; p-value<0.002).

– Our single cells transcriptomic analysis reveals that out of the genes differentially expressed between control and Dullard mutant cardiac NCC stand well known pan BMP signalling direct target genes, namely *Id1, Id2, Msx1* and *Msx2* (Figure 1F). Accordingly, *Id2* and *Msx2* detected using RNAscope probes exhibit elevated levels in the mutant NCC compared to control NCC or to adjacent positive tissues (Figure 1F). Hence, not only the phosphorylation of Smad1/5/8 is increased by the loss of Dullard, but their transcriptional activity is likely to be upregulated as well.

– We provide also data showing that the loss of Dullard using the *Pax3^cre^* driver not only lead to elevation of BMP signalling in cardiac NCC, but also in other lineages marked by this Cre driver and exposed to BMP ligands in their environment (Figure supplement 1D and data not shown). It includes for instance several muscle masses in the trunk and limb bud regions, as well as the dorsal neural tube. Furthermore, the muscle masses of Dullard mutants also display elevated levels of *Sema3c* (Figure 1—figure supplement 1D). These results further support the idea that Dullard stands as a pan and generic inhibitor of BMP signalling in several embryonic tissues.

Furthermore, we have attempted to rescue Dullard mutants by performing intraperitoneal injections to pregnant females with the Alk2 and Alk3 BMP receptors inhibitor LDN193189 at two different doses (Author response image 3). Unfortunately, we haven’t been able to downregulate the levels of PSmads neither in controls nor in Dullard mutants, and Dullard mutants treated with LDN still displayed OFT septation defect. This could stem from the instability in the LDN, issues from its absorption and diffusion within the mother and embryonic tissues.

**Author response image 3. respfig3:** LDN effect on Dullard mutants. (**A**) Analysis of phosho-Smad intensity at different levels of the OFT at E11.5 after 48h exposure to LDN. (**B**) Transversal sections of the distal OFT stained for PECAM and GFP after 48h treatment with LDN or vehicle of mutant E11.5 embryos.

We have also checked the state of TGFβ signalling in Dullard mutants, as this pathway has been shown to be increased upon deletion of the phosphatase in the limb bud mesenchyme (Hayata et al., 2015). Immunostaining for the phosphorylated form of one of the downstream effectors of this signalling pathway (Smad2) indicates that the levels of activation of this TF is not impaired in absence of Dullard (Figure 1—figure supplement 1E). Hence, in NCC Dullard does not appear to be required for dampening TGFβ signalling levels.

Finally, we have clarified in the Discussion our statement on whether the ingression of NCC may contribute to the generation of a BMP signalling gradient. From our data, it is clear that Dullard is not implicated in the establishment of the gradient of BMP signalling, but in the regulation of its amplitude. We have proposed that this is regulated by other negative regulators of the pathway and could be linked to temporal dynamics in this signalling pathway. Over time the signal decreases, hence cells which have been submitted earlier to BMP and which have ingressed further display lower levels of signalling than cells which have been in the presence of BMP ligands for a shorter period of time (Bier and Robertis, 2015).

Point 3. The possible role of Sema3C and other differentially expressed genes in the development of the phenotype.

As proposed by one of the reviewers, we have taken the opportunity to further develop our discussion in the revised version of the paper on how the differentially expressed genes may contribute to the establishment of the Dullard mutant phenotype. Notably, we believe that the elevation of *Sema3c* expression is central to the establishment of the OFT defects in Dullard mutants. Based on a handful of papers that have demonstrated in different cell lines Sema3c requirement for cohesion between cells, we propose that the elevated levels could promote the early compaction of the NCC in Dullard mutants.

References:

Bier E and Robertis EMD (2015) BMP gradients: A paradigm for morphogen-mediated developmental patterning. Science 348: aaa5838

Brown CB, Feiner L, Lu MM, Li J, Ma X, Webber AL, Jia L, Raper JA and Epstein JA (2001) PlexinA2 and semaphoring signaling during cardiac neural crest development. Dev. Camb. Engl. 128: 3071–3080

Hayata T, Yoichi, Ezura, Asashima M, Nishinakamura R and Noda M (2015) Dullard/Ctdnep1 Regulates Endochondral Ossification via Suppression of TGF-β Signaling. J. Bone Miner. Res. 30: 318–329

Jiang X, Rowitch DH, Soriano P, McMahon AP and Sucov HM (2000) Fate of the mammalian cardiac neural crest. Development 127: 1607–1616